EMBO
Molecular Medicine

# Follistatin is a novel therapeutic target and biomarker in FLT3/ITD acute myeloid leukemia

Bai-Liang He[1,2,†], Ning Yang[1,†], Cheuk Him Man[1], Nelson Ka-Lam Ng[1], Chae-Yin Cher[1], Ho-Ching Leung[1], Leo Lai-Hok Kan[1], Bowie Yik-Ling Cheng[1], Stephen Sze-Yuen Lam[1], Michelle Lu-Lu Wang[1], Chun-Xiao Zhang[1], Hin Kwok[3], Grace Cheng[3], Rakesh Sharma[3], Alvin Chun-Hang Ma[4], Chi-Wai Eric So[5] (iD), Yok-Lam Kwong[1] & Anskar Yu-Hung Leung[1,*] (iD)

## Abstract

Internal tandem duplication of Fms-like tyrosine kinase 3 (*FLT3*/ITD) occurs in about 30% of acute myeloid leukemia (AML) and is associated with poor response to conventional treatment and adverse outcome. Here, we reported that human *FLT3*/ITD expression led to axis duplication and dorsalization in about 50% of zebrafish embryos. The morphologic phenotype was accompanied by ectopic expression of a morphogen follistatin (*fst*) during early embryonic development. Increase in *fst* expression also occurred in adult *FLT3*/ITD-transgenic zebrafish, *Flt3*/ITD knock-in mice, and human *FLT3*/ITD AML cells. Overexpression of human *FST317* and *FST344* isoforms enhanced clonogenicity and leukemia engraftment in xenotransplantation model via *RET*, *IL2RA*, and *CCL5* upregulation. Specific targeting of *FST* by shRNA, CRISPR/Cas9, or antisense oligo inhibited leukemic growth *in vitro* and *in vivo*. Importantly, serum FST positively correlated with leukemia engraftment in *FLT3*/ITD AML patient-derived xenograft mice and leukemia blast percentage in primary AML patients. In *FLT3*/ITD AML patients treated with FLT3 inhibitor quizartinib, serum FST levels correlated with clinical response. These observations supported FST as a novel therapeutic target and biomarker in *FLT3*/ITD AML.

**Keywords** acute myeloid leukemia; follistatin; internal tandem duplication of Fms-like tyrosine kinase 3
**Subject Categories** Cancer; Haematology

## Introduction

Acute myeloid leukemia (AML) is characterized by an abnormal increase in myeloblasts in peripheral blood (PB) and bone marrow (BM). It is a heterogeneous disease with distinct clinicopathologic, cytogenetic, and genetic features in individual patients (Dohner *et al*, 2015). In young patients, intensive chemotherapy and allogeneic hematopoietic stem cell transplantation (HSCT) are the mainstays of treatment and 30–40% patients can achieve long-term remission (Papaemmanuil *et al*, 2016). In elderly patients not eligible for standard therapy, the outcome is poor.

Specific driver genetic mutations occur in different subtypes of AML, and therapies targeting mutations in AML have emerged: midostaurin (Stone *et al*, 2017) in combination with induction chemotherapy for upfront treatment and gilteritinib monotherapy (Perl *et al*, 2017) for relapsed or refractory disease in AML with FMS-like tyrosine kinase 3 (*FLT3*) mutations; and ivosidenib (DiNardo *et al*, 2018) and enasidenib (Stein *et al*, 2017) for relapsed or refractory AML with isocitrate dehydrogenase 1 and 2 (*IDH1 and IDH2*) mutations. Strategies targeting other genetic mutations and their activated cellular pathways are urgently needed (Lam *et al*, 2017).

Internal tandem duplication of *FLT3* (*FLT3*/ITD) occurs in about 30% of AML and is associated with poor treatment outcome (Patel *et al*, 2012; Smith *et al*, 2012). Constitutive activation of FLT3 arising from ITD activates downstream signals including PI3K/AKT, STAT5, and Ras/MEK/ERK (Mizuki *et al*, 2000; Martelli *et al*, 2006; Zhang *et al*, 2016). FLT3 inhibitors have been shown to be effective in *FLT3*/ITD AML (Assi & Ravandi, 2018). However, tyrosine kinase domain (TKD) mutations in *FLT3* confer drug resistance and are an important cause of treatment failure (Man *et al*, 2012; Smith *et al*, 2017). Identifying new pathogenetic signals in *FLT3*/ITD AML may result in novel therapeutic strategies.

1   Division of Hematology, Department of Medicine, Li Ka Shing Faculty of Medicine, The University of Hong Kong, Hong Kong SAR, China
2   Guangdong Provincial Key Laboratory of Biomedical Imaging, The Fifth Affiliated Hospital, Sun Yat-sen University, Zhuhai, Guangdong Province, China
3   Centre for Genomic Sciences, The University of Hong Kong, Hong Kong SAR, China
4   Department of Health Technology and Informatics, The Hong Kong Polytechnic University, Hong Kong SAR, China
5   Leukemia and Stem Cell Biology Group, Division of Cancer Studies, Department of Hematological Medicine, King's College London, London, UK
    *Corresponding author. Tel: +86 (852)22553347; Fax: +86 (852)29741165; E-mail: ayhleung@hku.hk
    †These authors contributed equally to this work

Follistatin (FST) was first identified from follicular fluid in the ovary and shown to suppress secretion of follicle-stimulating hormone from the anterior pituitary gland (Ueno et al, 1987). FST has been shown to express in different tissues (Phillips & de Kretser, 1998) and antagonizes activin A, a member of the TGF-β (transforming growth factor-β) superfamily (Cash et al, 2012). Alternative splicing of FST generates FST317 and FST344 that encode FST288 and FST315 proteins (Shimasaki et al, 1988), the latter being the predominant circulatory form. Fst had been reported to be regulated by CREB, FoxL2, and Smad3 in mouse gonadotrophic cell lines (Winters et al, 2007; Blount et al, 2009), by FoxO1 in mouse hepatocytes (Tao et al, 2018), and by Nrf2 in pulmonary epithelial cells (Lin et al, 2016). In mice, Fst knockout resulted in early postnatal mortality with multiple defects in muscles, skin, and bones (Matzuk et al, 1995). In zebrafish and Xenopus embryos, fst overexpression led to a dose-dependent dorsalization phenotype and, when ventrally expressed, induced a secondary body axis (Fainsod et al, 1997). In humans, FST has been implicated in the pathogeneses of prostate (Sepporta et al, 2013), ovarian (Di Simone et al, 1996), and liver (Rossmanith et al, 2002) cancers, and hyperglycemia (Tao et al, 2018). Mechanistically, both antagonism of TGF-β signaling and inhibition of cellular rRNA synthesis during glucose deprivation have been reported (Gao et al, 2010).

In this study, we demonstrated a hitherto undescribed pathogenetic role of FST in human AML with particular reference to FLT3/ITD. It originated from an unexpected finding that human FLT3/ITD induced ectopic fst expression during early embryonic development in zebrafish and caused axis duplication and dorsalization. Induction of FST expression by FLT3/ITD could be recapitulated in adult FLT3/ITD-transgenic zebrafish, Flt3/ITD knock-in mice, and human FLT3/ITD primary AML-derived murine xenografts, which was mediated by CREB phosphorylation. FST upregulated expression of RET, IL2RA, and CCL5 that collectively potentiated MAPK signaling and promoted leukemia growth in vitro and in vivo. FST targeting by shRNA, CRISPR/Cas9, and antisense oligo suppressed leukemia growth in vitro and in vivo. Serum FST correlated with clinical response to specific FLT3 inhibitor and subsequent leukemic progression in FLT3/ITD AML patients. These findings underscored the potential role of FST in AML as a new therapeutic target and biomarker during treatment with FLT3 inhibitors.

## Results

### Ectopic expression of FLT3/ITD induced axis duplication in zebrafish embryo

Previously, we demonstrated that expression of human FLT3/ITD by plasmid DNA injection in zebrafish embryos induced expansion of myelopoiesis reminiscent of the hematopoietic phenotype of Flt3/ITD knock-in mouse (He et al, 2014). In this study, FLT3/ITD was ectopically expressed in zebrafish embryos by mRNA injection. Intriguingly, axis duplication and dorsalization were observed in $15.2 \pm 1.3$ and $34.7 \pm 3.2\%$ of FLT3/ITD, but not FLT3/WT mRNA-injected embryos on 1 dpf (day post-fertilization) (Fig 1A–D; Appendix Fig S1A–J). This phenotype was not observed in FLT3/ITD plasmid DNA-injected embryos in our previous study (He et al, 2014). Axis duplication was confirmed by whole-mount in situ

hybridization (WISH) of notochord-specific marker col9a2 (Fig 1E–H). Constitutive phosphorylation and activation of FLT3 downstream signals STAT5, AKT, and ERK were confirmed in 293FT transfectant (Fig 1I) and zebrafish embryos (Fig 1J). Importantly, a specific FLT3 inhibitor quizartinib ameliorated the dorsalization and axis duplication anomalies in a dose-dependent fashion (Fig 1K), confirming the link between activation of flt3 signaling and the morphologic anomalies.

### FLT3/ITD upregulated FST expression in zebrafish and human AML

These developmental defects suggested disruption of normal morphogen gradients critical for dorsoventral (D-V) patterning and axis formation in vertebrates (De Robertis et al, 2000), including follistatin (fst) (Fainsod et al, 1997), goosecoid (gsc) (Steinbeisser et al, 1995), and chordin (chd) (Sasai et al, 1994). Therefore, their expression in FLT3/ITD-injected embryos was examined. The mRNA and protein expression of fst was significantly increased at shield stage (6 hpf) by 1.7- and 1.9-fold (Fig 1L and M). Ectopic expression of fst (Fig 1N) and goosecoid (gsc) was also shown by WISH (Appendix Fig S1K) and GFP reporter assay (Appendix Fig S1L–N). Consistently, upregulation of fst was also observed in FLT3/ITD plasmid DNA-injected embryos at 36 hpf, which could be effectively blocked by quizartinib treatment (Fig 2A–C). The relevance of fst to adult hematopoiesis was examined in transgenic zebrafish where human FLT3/ITD was expressed in hematopoietic stem cells (HSCs) (Fig 2D–K). FLT3/ITD expanded the myeloid and hematopoietic progenitor cell populations (Fig 2L–N) from whole kidney marrow (KM) (equivalence of mammalian BM) of adult transgenic zebrafish, where fst expression was significantly increased (Fig 2O).

In silico gene expression analysis based on BloodSpot database showed that total FST expression was upregulated in different cytogenetically defined AML subtypes relative to normal HSC (Appendix Fig S2A). Consistent with previous studies, isoform-specific RT–PCR showed that FST344 was the predominant FST transcript in FLT3/ITD AML cell lines MOLM-13 and MV4-11 (Appendix Fig S2B and C). Endogenous FST was expressed preferentially in the cytoplasm of FLT3/ITD-positive MOLM-13 cells at the perinuclear zone (Appendix Fig S2D). Total FST was significantly upregulated in primary myeloblasts obtained from FLT3/ITD AML patients relative to those in mobilized peripheral blood stem cells (PBSCs) from healthy donors (Fig 2P) and FLT3/WT AML samples (Appendix Fig S2E–G). Moreover, transfection of FLT3/ITD led to a significant upregulation of total FST in HeLa cells (Appendix Fig S2H).

### FST is a CREB target gene in FLT3/ITD AML

In silico analysis of transcription factor (TF) binding sites in FST promoter was performed. Binding sites for cAMP-response element binding protein (CREB) are over-represented (Fig 3A). Direct binding of p-CREB to FST promoter in FLT3/ITD-positive MOLM-13 cells was confirmed by ChIP-qPCR. Compared with normal IgG control, there was a sixfold increase in DNA binding to p-CREB in FST promoter (Fig 3B and C). Dual-luciferase reporter assay demonstrated that deletion of CREB-binding site on FST promoter

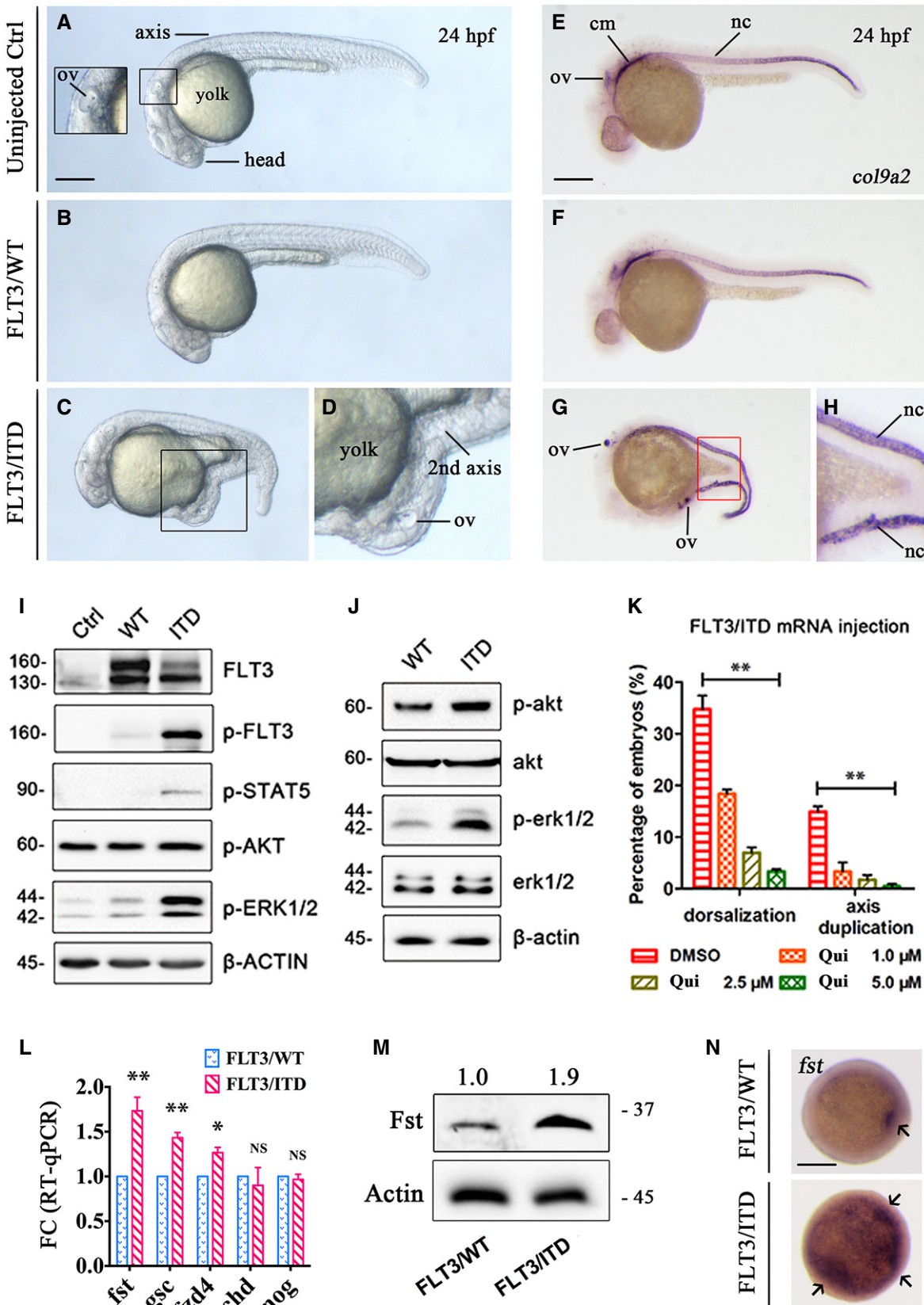

**Figure 1.**

**Figure 1.  Overexpression of FLT3/ITD induced axis duplication and ectopic expression of FST in zebrafish embryos.**

A–D   The morphology of uninjected, *FLT3*/WT mRNA, and *FLT3*/ITD mRNA-injected (150 ng per embryo) embryos on day 2 post-fertilization (dpf).

E–H   Whole-mount *in situ* hybridization (WISH) of notochord-specific marker *col9a2* in uninjected, *FLT3*/WT mRNA, and *FLT3*/ITD mRNA-injected embryos on 2 dpf.

I, J   FLT3 signaling was detected by Western blotting in 293FT cells transfected with *FLT3*/ITD mRNA (I) or in zebrafish embryos injected with *FLT3*/ITD mRNA (J).

K   The effect of FLT3 inhibitor quizartinib (Qui) on the dorsalization and axis duplication phenotype induced by *FLT3*/ITD mRNA injection in zebrafish.

L–N   Quantification of *fst* expression by RT–qPCR (L), Western blotting (M), and WISH (N) after *FLT3*/ITD overexpression in zebrafish embryos at 6 hpf.

Data information: ov: otic vesicles; cm: cephalic mesoderm; nc: notochord. Scale bar = 500 μm. In (K and L), the experiments were performed in triplicates and the data are presented as mean ± SEM. *$P < 0.05$ and **$P < 0.01$ (Student's *t*-test). NS, not significant.

Source data are available online for this figure.

abolished the effect of CREB-mediated FST upregulation (Fig 3D). Isogenic Ba/F3 cells transduced with *FLT3*/ITD was used to test the regulation between *FLT3*/ITD and *FST*. Consistently, FLT3/ITD, its downstream signals including p-STAT5, p-ERK1/2, p-AKT, and p-4E-BP1, and FST expression were increased in Ba/F3-*FLT3*/ITD cells compared to the parental cells (Fig 3E). CREB phosphorylation was also significantly increased (Fig 3F). Quizartinib that suppressed FLT3 signaling (Appendix Fig S2I) reduced CREB phosphorylation (Fig 3G), *Fst* transcription, and expression (Fig 3H) in Ba/F3-*FLT3*/ITD cells. CREB had been reported to be activated by p90RSK (a.k.a. RSK1) (Sakamoto & Frank, 2009), which is a known downstream effector of *FLT3*/ITD-ERK signaling cascade (Elf *et al*, 2011). Indeed, inhibition of p90RSK by BRD7389 effectively reduced FST expression and phosphorylation of CREB in MOLM-13 (Fig 3I and J) and Ba/F3-*FLT3*/ITD cells (Fig 3J and K). Consistently, a doxycycline-inducible system showed that knockout of *p90RSK* by CRISPR/Cas9 resulted in significant decrease in FST expression in MOLM-13 cell line (Fig 3L). Specifically, CREB inhibitor 666-15 (Kang *et al*, 2015) or knockout by CRISPR/Cas9 also decreased FST expression (Fig 3M and N) and CREB inhibitor 666-15 reduced cell growth of Ba/F3-*FLT3*/ITD without supplemental IL-3, but not Ba/F3 parental cells or Ba/F3-*FLT3*/ITD cells supplemented by IL-3 (Fig 3O). Furthermore, 666-15 treatment partially rescued the morphologic anomalies induced by *FLT3*/ITD in zebrafish embryos (Fig 3P) without observable toxicity, underscoring the preferential role of CREB in *FLT3*/ITD signaling. These observations demonstrated the presence of a FLT3-p90RSK-CREB-FST signaling axis that might be relevant in the pathogenesis of *FLT3*/ITD.

## FST potentiates MAPK/ERK signaling to promote leukemia cell growth

To delineate the pathogenetic roles of FST in AML, *FST* was overexpressed in the AML line ML-2, which showed the lowest endogenous FST expression (Fig 4A). Overexpression of the two spliced forms of FST (*FST317* and *FST344*) was confirmed at mRNA and protein levels (Fig 4B) and was shown to enhance cell growth (Fig 4C) and clonogenicity (Fig 4D and E) *in vitro*, respectively. Transplantation of ML-2 cells overexpressing either of these spliced variants into NSG mice demonstrated increased leukemia engraftment (Fig 4F and G) and shortened animal survival (Fig 4H).

To identify the underlying molecular targets of FST, changes in gene expression profiles upon *FST344* expression in ML-2 cells were examined (Table EV1). Upregulation of genes involved in signal transduction, early endosome formation, mitogen-activated protein kinase (MAPK) cascade (*RET, IL2RA, CCL5*), protein kinase activity, and cell proliferation are shown (Fig 4I; Appendix Fig S3A and B). Upregulation of *RET, IL2RA*, and *CCL5* was confirmed by RT–qPCR (Fig 4J), of which *IL2RA* and *CCL5* were shown to highly express in human AML compared with normal hematopoietic tissues (Appendix Fig S4) and were associated with poor overall survival in AML patients (Fig 4K and L). In fact, activation of ERK1/2 and upregulation of its downstream effector RSK1 were demonstrated in ML-2 cells overexpressing *FST* (Fig 4M). Similarly, *FST* overexpression in MOLM-13 cells (Appendix Fig S3C), which showed intermediate *FST* expression, also potentiated cell growth (Appendix Fig S3D) and MAPK/ERK activation (Appendix Fig S3E). Genes

**Figure 2.   FST was increased in FLT3/ITD-transgenic zebrafish and FLT3/ITD-mutated AML.**

A–C   WISH of *fst* in *FLT3*/WT (A), and *FLT3*/ITD plasmid DNA-injected zebrafish embryos without (B) or with (C) quizartinib treatment (2.5 μM) from 6 to 36 hpf. *fst* expression was expanded by *FLT3*/ITD DNA in 86% of embryos (B, arrow, 32/37) which could be effectively blocked by treating with FLT3 inhibitor quizartinib in 83% of embryos (C, 29/35).

D–K   Generation and characterization of *FLT3*/ITD-transgenic zebrafish. Diagrammatic representation (D and E) of the generation of Runx1-*FLT3*/ITD-transgenic zebrafish (see Materials and Methods section). GFP expression was detected by fluorescent microscopy (F–H) and in blood circulation and thymus by WISH (I and J, blue arrow) in WT sibling and Runx1-*FLT3*/ITD-transgenic zebrafish (F1) embryos at 4 dpf. *FLT3*/ITD-positive zebrafish (F1) were confirmed by PCR genotyping of *GFP* and *FLT3*/ITD using genomic DNA from fin clip of WT siblings and Runx1-*FLT3*/ITD-transgenic zebrafish (F1) at 2 months old. Fish 4, 5, and 6 showed germline transmission of *FLT3*/ITD transgene (K).

L–N   Kidney marrow (KM) was collected from Runx1-*FLT3*/ITD-transgenic zebrafish (F1) at 18 months old. The morphology and hematopoietic composition of KM from WT siblings ($n = 6$) and Runx1-*FLT3*/ITD-transgenic ($n = 6$) zebrafish were examined by Giemsa staining (L) and flow cytometry (M, N) (abbreviation for panel M: M, myeloid cells; P, progenitor cells; L, lymphoid cells; E, erythroid cells). Data are presented in box plot. The whiskers, boxes, and central lines in panel N represented the minimum-to-maximum values, 25th-to-75th percentile, and the 50th percentile (median), respectively. **$P < 0.01$ (Student's *t*-test).

O   Expression of *fst* was detected by RT–qPCR in KM from WT sibling and Runx1-*FLT3*/ITD-transgenic zebrafish at 18 months old. The RT–qPCR experiments were performed in triplicates, and data were presented as mean ± SEM. **$P < 0.01$.

P   Detection of FST expression, p-ERK1/2, and p-CREB in mononuclear cells from normal peripheral blood stem cell (PBSC) and *FLT3*/ITD AML patients (diagnostic samples with leukemia blasts > 80%) by Western blotting. ^: non-specific staining of p-ATF1 protein due to the conserved motif.

Data information: Scale bar = 500 μm.

Source data are available online for this figure.

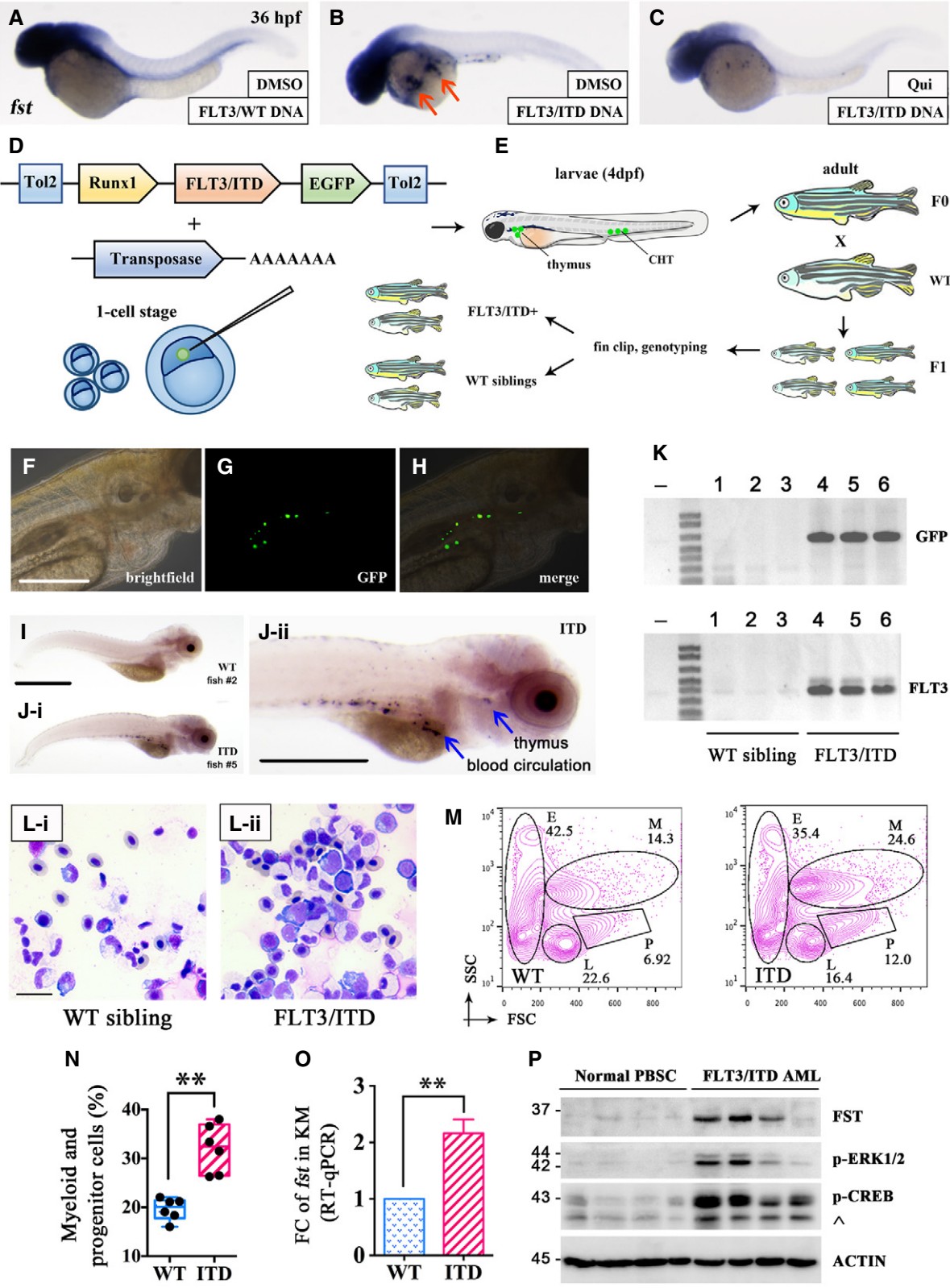

**Figure 2.**

associated with cell adhesion, oxidative stress response, differentiation, immune response, negative regulation of FGF receptor signaling pathway, and extracellular matrix organization were

downregulated after *FST* overexpression (Appendix Fig S3F). RNA-seq was also performed in ML-2 cell-overexpressed *FST317* compared to those with GFP overexpression. Intriguingly, the

transcriptome profile induced by *FST317* overexpression was distinct from that induced by *FST344* overexpression (Appendix Fig S5A, Table EV2). Specifically, the differentially expressed gene RAS oncogene family like 6 (*RABL6*, also known as *RBEL1* and *C9orf86*) (Hagen *et al*, 2014), *CD93* (Iwasaki *et al*, 2015), and Zinc Finger Protein 709 (*ZNF709*) (Yan *et al*, 2016) have been implicated in

cancers and leukemia. Upregulation of *PRTFDC1* and *PODXL2* expression was correlated with poor survival of AML, whereas downregulation of *CCNL1* and *RP11-762I7.5* was correlated with poor survival of AML (Appendix Fig S5B).

Proteomics analyses in both *FST344-* and *FST317-*overexpressing ML-2 cells showed consistent increases in protein expression of

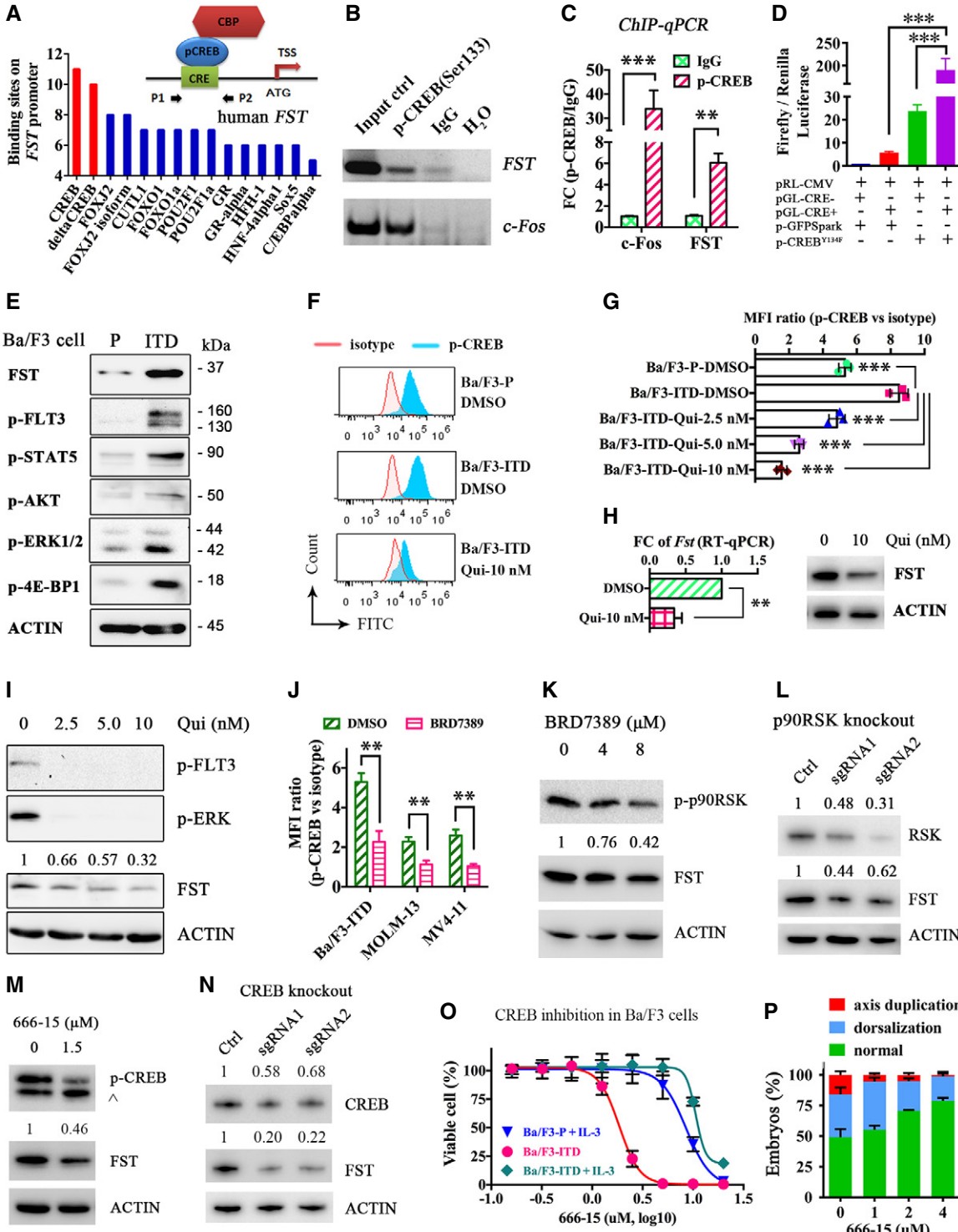

**Figure 3.**

**Figure 3.  *FLT3*/ITD upregulated *FST* through phosphorylation of CREB.**

A   *In silico* analysis (DECipherment of DNA Elements, SABiosciences) and schematic model of transcription factor binding sites on human *FST* promoter. CBP: CREB-binding protein; CRE: cAMP-response element; TSS: transcription start site.

B, C   The direct binding of p-CREB to human *FST* promoter was detected by ChIP-PCR (B) and ChIP-qPCR (C). *c-Fos* was used as positive control of p-CREB target gene. Normal IgG was used as negative control of ChIP.

D   Dual-luciferase assay demonstrating the direct binding of p-CREB on human FST promoter. pRL-CMV, Renilla luciferase vector; pGL-CRE− and pGL-CRE+, firefly luciferase expression driven by human *FST* promoter with deleted CRE site (CRE−) or wild type (CRE+); p-GFPSpark, GFP-expressing vector; p-CREB$^{Y134F}$, CREB$^{Y134F}$-GFP-expressing vector.

E   FST expression and *FLT3*/ITD signaling were detected by Western blotting in Ba/F3-parental (P in short) and Ba/F3-*FLT3*/ITD (ITD in short) cells.

F–H   Phospho-flow analysis of p-CREB in Ba/F3-parental, Ba/F3-*FLT3*/ITD, and Ba/F3-*FLT3*/ITD cells treated with FLT3 inhibitor quizartinib (Qui in short). Isotype antibody was used as control to calculate the mean fluorescence intensity (MFI) ratio (F, G). The transcription and expression of *Fst* were detected by RT–qPCR after quizartinib treatment (10 nM) in Ba/F3-*FLT3*/ITD cells for 1 day (H).

I–K   The expression of FST and phosphorylation of CREB were detected by Western blotting (I and K) and phospho-flow analysis (J) in MOLM-13 (I) and Ba/F3-*FLT3*/ITD (K) cells treated with quizartinib and BRD7389 for 1 day, respectively.

L   RSK expression and FST expression were detected by Western blotting after p90RSK knockout by CRISPR/Cas9 in MOLM-13 cells.

M   The phosphorylation of CREB and FST expression was detected by Western blotting in Ba/F3-*FLT3*/ITD cells treated with CREB inhibitor 666-15 for 1 day. ^: non-specific staining of p-ATF1 protein due to the conserved motif.

N   CREB expression and FST expression were detected by Western blotting after CREB knockout by CRISPR/Cas9 in MOLM-13 cells.

O   The growth of Ba/F3-parental (with IL-3), Ba/F3-*FLT3*/ITD (without IL-3), and Ba/F3-*FLT3*/ITD (with IL-3) cells was measured after 3 days treatment of CREB inhibitor 666-15 *in vitro*.

P   The rescue effect of CREB inhibitor 666-15 on *FLT3*/ITD-induced dorsalization and axis duplication in zebrafish embryos at 1 dpf.

Data information: In (C, D, G, H, J, and O), the experiments were performed in triplicates, and the data were presented as mean ± SEM. **$P < 0.01$ and ***$P < 0.001$ (Student's *t*-test).
Source data are available online for this figure.

Cathepsin G (CTSG), TATA-box binding protein-associated factor 15 (TAF15), CD44, RNA binding motif protein 39 (RBM39), hematological and neurological expressed 1 (HN1), and Poly(C)-binding protein 2 (PCBP2) (Appendix Fig S6A and B; Table EV3) of which TAF15 (Ballarino *et al*, 2013), CD44 (Quere *et al*, 2011), HN1 (Zhang *et al*, 2017), and PCBP2 (Han *et al*, 2013) have been associated with cancer initiation and progression. On the other hand, proteins associated with nonsense-mediated decay of mRNA (NMD) were significantly decreased in *FST344*-overexpressing ML-2 cells (Appendix Fig S6C). As FST is an antagonist of activin during embryonic development (Fainsod *et al*, 1997) and cancer pathogenesis (Gao *et al*, 2010), we examined the expression and role of activin in AML cell lines. In MOLM-13 and MV4-11 cells, activin A and C and their receptors were expressed (Appendix Fig S7A). However, exogenous FST (Appendix Fig S7B) and activin A (Appendix Fig S7C), and activin receptor antagonist (Appendix Fig S7D) and FST-neutralizing antibody (Appendix Fig S7E) had no significant effect on MOLM-13 cell growth *in vitro*, enumerated as the number of viable cells based on PrestoBlue assay. Therefore, *FLT3*/ITD-mediated FST upregulation enhanced cellular proliferation likely

by intracellular signaling rather than an autocrine/paracrine mechanism involving activin antagonism.

## Targeting FST for AML treatment

In MOLM-13 cell line, *FST* knockdown (Fig 5A) induced apoptosis (Fig 5B and C), significantly reduced colony formation (Fig 5D and E), and reduced engraftment (Fig 5F and G) and prolonged survival of MOLM-13 cell-engrafted NSG mice (Fig 5H). Similarly, *FST* knockdown also reduced clonogenicity of another FLT3/ITD-mutated MV4-11 AML cells *in vitro* (Appendix Fig S8A–C). Consistently, *FST* targeting by CRISPR/Cas9 (Appendix Fig S8D and E) in MOLM-13 cells similarly reduced its clonogenicity (Fig 6A and B) and engraftment into NSG mice (Fig 6C and D) and prolonged survival of engrafted animals (Fig 6E).

As a proof of principle, effects of a cell-permeant antisense oligo (ASO) against human *FST* were examined to evaluate the clinical relevance of *FST* as a target of therapeutic intervention. FST-ASO3 significantly reduced *FST* transcription (Fig 6F) and inhibited leukemia growth of MOLM-13 cells *in vitro* (Fig 6G). Intraperitoneal

**Figure 4.  FST promoted leukemia growth by activating ERK.**

A   FST expression in different AML cell lines was detected by Western blotting.

B, C   *FST317* and *FST344* overexpression resulted in significant increases in *FST* transcription by RT–qPCR and protein by Western blot (B) and promoted ML-2 cell growth *in vitro* (C). Green, ML-2-GFP; blue, ML-2-FST317; red, ML-2-FST344. The RT–qPCR experiments were performed in triplicates (B).

D, E   The clonogenicity of ML-2 overexpressing *GFP*, *FST317*, and *FST344 in vitro* for 14 days. The CFU experiments were performed in triplicates (E).

F–H   The engraftment of ML-2 (with luciferase gene) overexpressing *GFP*, *FST317*, and *FST344* was quantified by bioluminescence imaging (F and G), and the survival of ML-2-engrafted NSG mice *in vivo* was recorded (H). Survival curve in panel H was analyzed by log-rank test. *$P < 0.05$ and **$P < 0.01$.

I, J   RNA-seq and RT–qPCR validation of upregulation of *RET*, *IL2RA*, and *CCL5* after *FST344* overexpression in ML-2 cells. RT–qPCR experiments were performed in triplicates (J).

K, L   Overall survival analysis of patients from TCGA-AML based on the differential expression of *IL2RA* and *CCL5*.

M   Activation of MAPK/ERK pathway in ML-2 by *FST344* overexpression was validated by Western blotting. Scale bar = 1 mm.

Data information: In (B, C, E, G, and J), data were presented as mean ± SEM. *$P < 0.05$, **$P < 0.01$, and ***$P < 0.001$ (Student's *t*-test). ns: not significant.
Source data are available online for this figure.

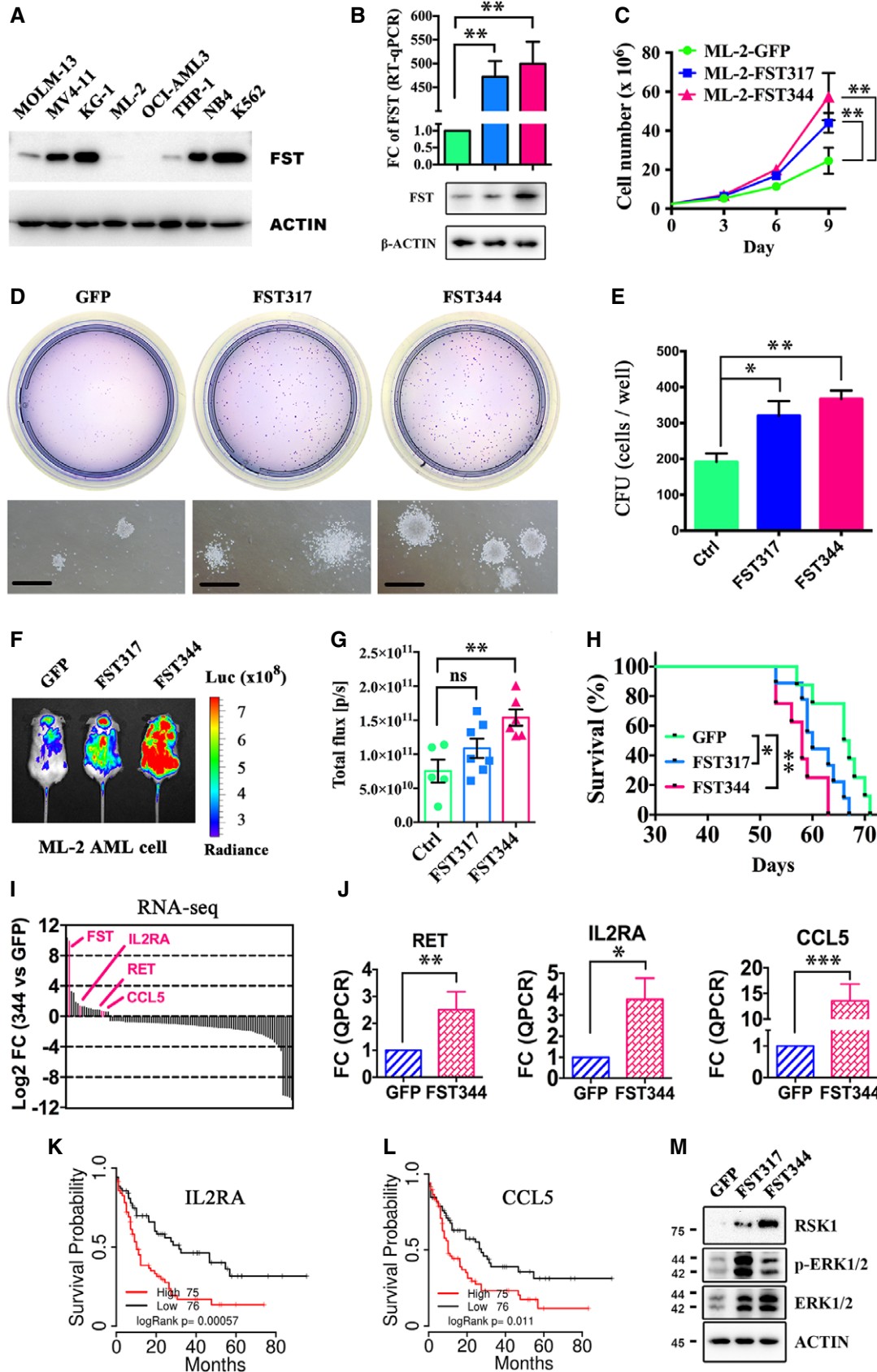

Figure 4.

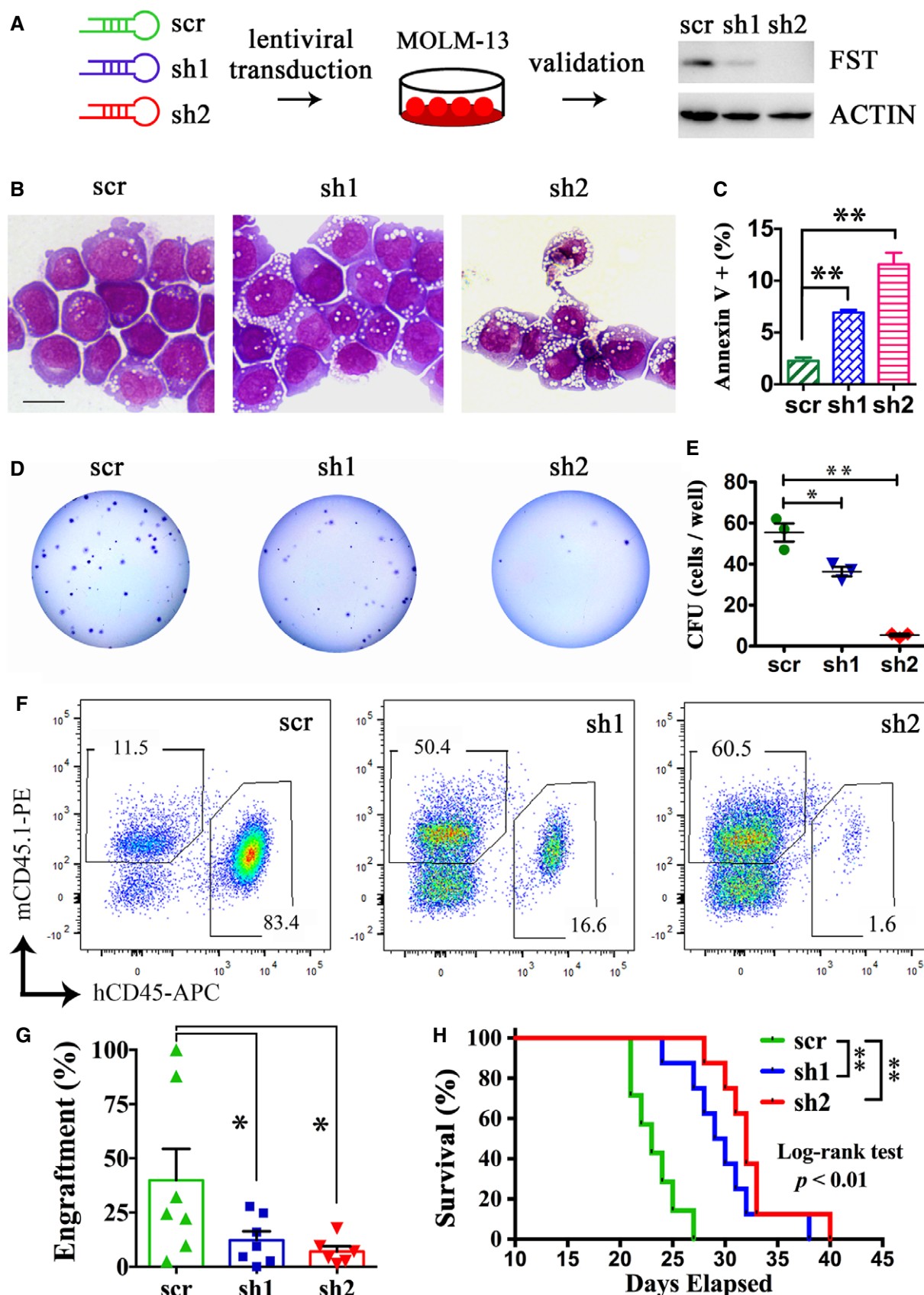

Figure 5.

**Figure 5.** *FST* knockdown reduced AML cell growth *in vitro* and *in vivo*.

A    *FST* knockdown in MOLM-13 by shRNA effectively reduced FST expression.

B–E  The morphology (B), apoptosis (C), and clonogenicity of MOLM-13 (D and E) were measured after *FST* knockdown *in vitro*. Scale bar = 10 μm. The apoptosis assays (C) were performed in triplicates.

F, G  The engraftment of MOLM-13 after *FST* knockdown was detected by flow cytometry of human CD45- and mouse CD45.1-positive cells in recipient mouse BM aspiration at week 2 post-transplantation.

H    The effect of *FST* knockdown on the survival of NSG mice engrafted with MOLM-13. scr: scrambled sequence control (7 mice); sh: short hairpin RNA (8 mice for sh1 and sh2, respectively). The survival curve was analyzed by log-rank test. **$P < 0.01$.

Data information: In (C, E, and G), data were presented as mean $\pm$ SEM. *$P < 0.05$ and **$P < 0.01$ (Student's *t*-test).

Source data are available online for this figure.

injection of FST-ASO3 significantly prolonged the survival of MOLM-13-engrafted NSG mice (Fig 6H). These observations supported the proposition that *FST* is a potential therapeutic target for the treatment of *FLT3*/ITD AML.

### FST as a biomarker for leukemia progression and therapeutic response

Serum FST was evaluated for its potential as a biomarker for monitoring of AML progression. Cell lines including ML-2 and MOLM-13 overexpressing FST showed higher levels of FST in the culture medium (Appendix Fig S9), suggesting that in addition to its role in intracellular signaling, FST was also secreted. A higher serum Fst level could also be demonstrated in *Flt3*/ITD knock-in mice compared with their wild-type siblings (Fig 7A–D). In addition, serum FST levels were significantly increased after engraftment of MOLM-13 cells in NSG mice (Fig 7E–G), which could be reduced after shRNA-mediated *FST* knockdown *in vivo* (Fig 7H). Furthermore, serum FST levels from *FLT3*/ITD AML patient-derived xenograft mice were significantly increased at week 6 post-xenotransplantation (Fig 7I–K). Importantly, serum FST levels were positively correlated with the percentage of leukemia blast in PB from FLT3/ITD-mutated primary AML patients at diagnosis (Fig 7L). Moreover, in patients with relapsed or refractory *FLT3*/ITD AML treated with quizartinib monotherapy, serum FST levels were significantly decreased in four patients upon morphological clearance of marrow blasts at CRi (complete remission with incomplete hematological recovery) but resurged during relapse (Fig 7M). In one patient who did not respond to treatment, serum FST level continued to rise throughout the course of treatment (Fig 7N). Serum FST response upon remission was specific to FLT3/ITD AML and was not seen in *FLT3*/WT AML patients who achieved complete

remission with conventional chemotherapy (Appendix Fig S10). However, serum FST levels in different AML patients varied widely and there was no convincing difference between FLT3-ITD and FLT3-WT AML (Appendix Fig S11).

## Discussion

In this study, we demonstrated a hitherto undescribed FLT3/ITD-ERK-p90RSK-CREB-FST signaling axis relevant in *FLT3*/ITD AML. The serendipitous axis duplication and dorsalization in zebrafish embryos were not detected in our previous studies (He *et al*, 2014). The apparent discrepancies in morphologic phenotypes were related to the difference in temporal and spatial expression of nucleic acids injected. In our previous study, overexpression of FLT3/ITD by plasmid DNA injection resulted in relatively late (after 6 h post-fertilization) and mosaic expression of FLT3/ITD that had minimal effects on dorsal–ventral patterning. In the present study, injection of FLT3/ITD mRNA into one-cell stage zebrafish embryos resulted in early and ubiquitous expression of FLT3/ITD proteins that induced axis duplication and dorsalization in embryos. STAT5, AKT, and ERK inhibitors were associated with significant embryonic toxicities, making it difficult to ascertain their effects on dorsalization or axis duplication induced by *FLT3*/ITD expression. Mechanistically, FST enhanced MAPK/ERK signaling, providing a positive feedback loop that promoted leukemia cell growth *in vitro* and *in vivo*. Perturbation of *FST* expression significantly reduced leukemia clonogenicity and engraftment *in vivo*. Serum FST levels were significantly increased in *Flt3*/ITD knock-in mouse and *FLT3*/ITD AML patient-derived xenografted mice and were correlated with clinical responses to quizartinib in AML patients. Collectively, our data provided important insights into the oncogenic role of *FST* in AML

**Figure 6.** *FST* targeting by CRISPR/Cas9 or antisense oligo significantly reduced leukemia cell growth *in vitro* and *in vivo*.

A, B  The morphology and clonogenicity of MOLM-13 after *FST* knockout by CRISPR/Cas9 *in vitro*. Scale bar = 10 μm.

C, D  The engraftment of MOLM-13 after *FST* knockout was detected by flow cytometry of human CD45- and mouse CD45.1-positive cells in recipient mouse BM aspiration at week 2 post-transplantation.

E    The effect of *FST* knockout on the survival of NSG mice engrafted with MOLM-13 cells. Cas9: Cas9 only (10 mice); sgRNA#3: Cas9 + sgRNA#3 (10 mice); sgRNA#4: Cas9 + sgRNA#4 (8 mice).

F    The knockdown efficiency of different *FST*-specific antisense oligos (ASOs) in MOLM-13 cells was detected by RT–qPCR after 3 days of treatment *in vitro*. The knockdown and RT–qPCR experiments were performed in triplicates.

G    MOLM-13 cell growth was measured after 3 days of treatment of FST-ASO *in vitro*. The ASO treatment experiments were performed in triplicates.

H    Intraperitoneal injection of *FST*-ASO (10 mg/kg weekly, 6 mice) significantly prolonged the survival of MOLM-13-engrafted NSG mice. The random sequence was used for negative control (Neg-ASO, 6 mice).

Data information: In (B, D, F, and G), data were presented as mean $\pm$ SEM. *$P < 0.05$ and **$P < 0.01$ (Student's *t*-test). In (E and H), survival curves were analyzed by log-rank test. *$P < 0.05$.

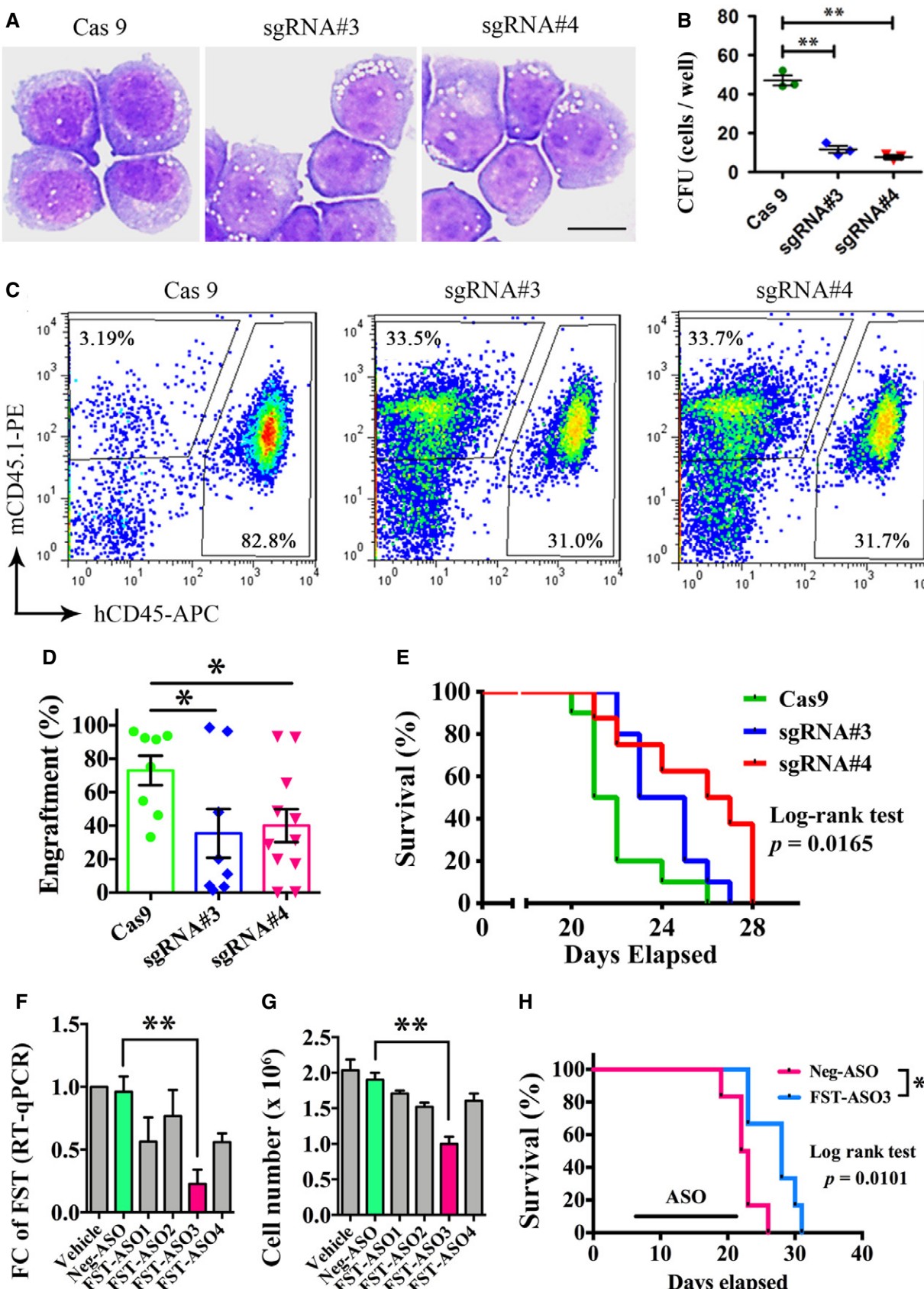

**Figure 6.**

and suggested a new therapeutic target and biomarker in AML treatment (Fig 8).

Firstly, *FLT3*/ITD was shown to upregulate *FST* expression by activating CREB in zebrafish and human AML. As a transcription factor, CREB is activated through phosphorylation by various stress and cytokine stimuli that increase intracellular cyclic AMP or calcium. When activated, CREB dimerizes and binds to promoter region of target genes that contain CRE sites. It is involved in various cellular processes including anti-apoptosis, immune regulation, cell cycle progression, and cytokine signal transduction (Cheng *et al*, 2008). As small molecule inhibitors targeting CREB are being developed (Mitton *et al*, 2016), the involvement of CREB in *FLT3*/ITD pathogenesis as demonstrated in this study may provide new opportunities for therapeutic combination with FLT3 inhibitors in *FLT3*/ITD AML.

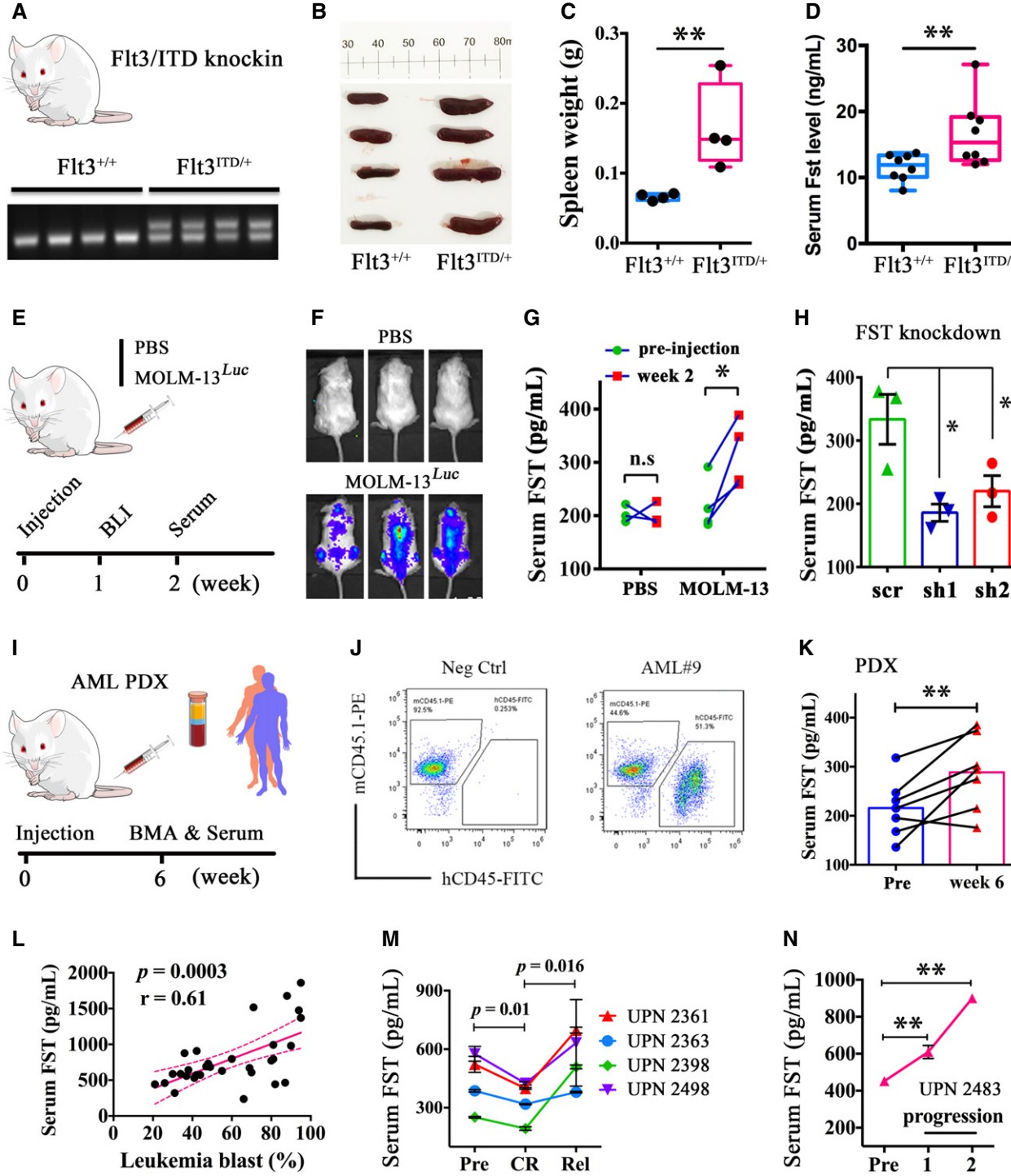

Figure 7.

**Figure 7. Serum FST correlated with leukemia progression and therapeutic response in mouse and human *FLT3*/ITD AML.**

A–D *Flt3*/ITD knock-in mouse were genotyped (A). The spleen weight (B and C, 4 mice each) and serum Fst level (D) were measured in WT siblings and *Flt3*/ITD knock-in mice (8 mice each). In (C and D), data were presented in box plot. The whiskers, boxes, and central lines represented the minimum-to-maximum values, 25th-to-75th percentile, and the 50th percentile (median), respectively. **$P < 0.01$ (Student's *t*-test).

E–H The engraftment of MOLM-13 in NSG mice was confirmed at week 1 post-injection (E, F). Serum FST level was measured in MOLM-13-engrafted NSG mice at pre-injection and week 2 post-injection (G, PBS group, 3 mice; MOLM-13 group, 4 mice). After *FST* knockdown, serum FST level was also measured in MOLM-13-engrafted NSG mice at week 2 post-injection (H, 3 mice for each group).

I–K Serum FST level was significantly increased in primary AML-derived xenografted mouse at week 6 post-injection. Human primary AML cells (*FLT3*/ITD-positive, leukemia blasts > 80%, $10 \times 10^6$, $n = 7$) were injected via tail vein into irradiated NSG mice at 6–8 weeks old (I). Human leukemic engraftments were confirmed by flow cytometry of human CD45 and mouse CD45.1 cells in recipient mouse BM aspiration (J). Serum FST from pre-injection and post-engraftment mouse was measured (K, one mouse for each primary AML sample).

L Correlation between serum FST levels and leukemia blast percentage from FLT3/ITD-mutated AML at diagnosis. Correlation analysis (Pearson's correlation coefficient) was performed by GraphPad Prism 6.

M Serum FST decreased in CR and increased after relapse in 4 AML patients receiving quizartinib monotherapy.

N Serum FST continued to rise during disease progression from a patient who did not respond to quizartinib. Patients in (M) and (N) were recruited in the QUANTUM-R, and patient accrual has been completed.

Data information: In (G and K), data were presented as scatter dot plot. n.s: not significant, *$P < 0.05$ and **$P < 0.01$ (Student's *t*-test). In (H, M and N), data were presented as mean ± SEM. *$P < 0.05$ and **$P < 0.01$ (Student's *t*-test).

Secondly, we demonstrated a novel pathogenetic link between CREB and FST in human *FLT3*/ITD AML. As a prototype transcription factor, CREB is overexpressed and constitutively phosphorylated in different types of human cancers (Shankar *et al*, 2005), including AML, and plays critical role in leukemogenesis (Sakamoto & Frank, 2009). The activated CREB regulates transcription of a number of targeted genes, including *cyclin* (Desdouets *et al*, 1995), *Bcl-2* family members (Wilson *et al*, 1996), *Egr-1* (Mayer *et al*, 2008), and *c-Fos* (Ginty *et al*, 1994). In the present study, we showed that CREB binds on *FST* promoter directly in *FLT3*/ITD AML cells. FST expression was suppressed by pharmacologic inhibition of CREB both in an isogenic Ba/F3-*FLT3*/ITD cell model and in

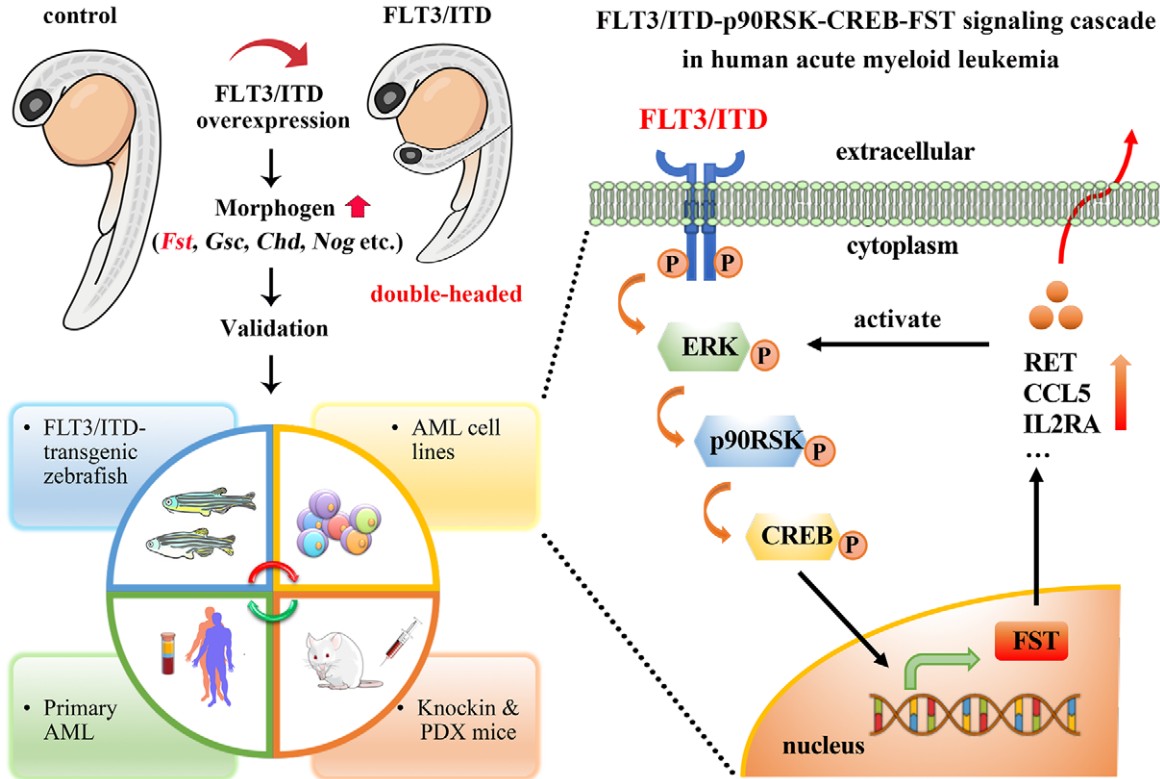

**Figure 8. Schematic diagram depicting the molecular mechanism of FST upregulation in FLT3/ITD-transgenic zebrafish, knock-in mice, and human AML.**

*Left panel*. Transient overexpression of human FLT3/ITD mutation resulted in axis duplication and dorsalization abnormalities in zebrafish accompanied by upregulation of embryonic morphogen Fst. Upregulation of FST was consistently found in FLT3/ITD-transgenic zebrafish, Flt3/ITD knock-in mice, FLT3/ITD-mutated AML cell lines, and primary AML samples *in vitro* and *in vivo*. *Right panel*. A novel FLT3/ITD-p90RSK-CREB-FST signaling cascade was demonstrated in human AML. FST is a promising biomarker and therapeutic target for human FLT3/ITD+ AML.

*FLT3*/ITD AML patients treated with a specific FLT3 inhibitor quizartinib. p90RSK and CREB knockout by CRISPR/Cas9 also induced significant decrease in FST expression.

Thirdly, transcriptome analyses upon *FST* overexpression have provided insights about downstream targets of FST, some of which were of relevance to the pathogenesis of AML and could generate testable hypotheses. Specifically, *FST* overexpression in ML-2 cells was shown to upregulate genes involved in MAPK/ERK pathway including *RET*, *IL2RA*, and *CCL5*, of which *IL2RA* and *CCL5* were shown to highly express in human AML compared with normal hematopoietic tissues (Bloodspot website: http://servers.binf.ku.d k/bloodspot/) and were associated with poor overall survival in AML patients. Our observations might provide an explanation for a higher *IL2RA* (*CD25*) expression among *FLT3*/ITD myeloblasts, as well as its correlation with *FLT3*/ITD allelic burden (Abd El-Ghaffar *et al*, 2016), and corroborated with recent findings of CD25 as one of the leukemia-associated immunophenotypes (LAIP) specific for *FLT3*/ITD AML (Angelini *et al*, 2015). CCL5 (RANTES) is a chemokine released from human AML cells and was thought to induce T-cell chemotaxis (Olsnes *et al*, 2006). Its pathogenetic link with FLT3/ITD and FST and its potential in development of immunotherapy would have to be examined in the future. Recently, the *RET* proto-oncogene was shown to be upregulated in AML and might promote leukemogenesis through inhibition of mTORC1-regulated autophagy (Rudat *et al*, 2018), stabilizing oncogenic proteins including *FLT3*/ITD that are susceptible to autophagic degradation. Upregulation of *RET* expression by *FST* might result in a positive feedback loop and sustain the oncogenic signals of *FLT3*/ITD.

The proteomics studies have also generated information that might become important leads for future research. Overexpression of *FST344* and *FST317* in ML-2 cells consistently upregulated proteins with oncogenic properties. TAF15 is a nuclear protein shown to induce rapid cellular proliferation through regulation of miRNA (Ballarino *et al*, 2013). HN1 expression was upregulated in human cancers, which might promote oncogenesis MYC (Zhang *et al*, 2017). PCBP2 expression was associated with poor prognosis in glioblastoma (Luo & Zhuang, 2017), and its overexpression increased colony formation and invasion capability (Han *et al*, 2013). CD44 mediates cell–cell or cell–extracellular matrix interactions, and its blockade with a monoclonal antibody had been shown to revert differentiation block of myeloblasts, inhibit their proliferation (Charrad *et al*, 2002), and block leukemia stem cells (LSC) from trafficking to their supportive niche (Jin *et al*, 2006). These new targets in FLT3/ITD will provide important foundation for the development of new therapeutic strategies in this AML subtype.

Results of the present study are of clinical relevance. Specifically, perturbation of *FST* expression by shRNA and CRISPR/Cas9 ameliorated leukemogenesis. Importantly, antisense oligo targeting FST also reduced leukemia growth *in vitro* and *in vivo*, providing an important lead for clinical trials. Furthermore, a multiprong approach targeting FLT3/ITD, FST, and other molecules including CREB and RET proto-oncogene may improve treatment outcome of *FLT3*/ITD AML. Finally, plasma FST levels might be exploited as a surrogate biomarker of leukemia cell growth to evaluate treatment response to FLT3 inhibitors in *FLT3*/ITD AML. Despite its ubiquitous tissue expression, the observations aforementioned supported the proposition that FLT3-ITD AML blasts contributed, at least partially, to plasma FST. Intriguingly, FST-neutralizing antibodies had no significant effect on leukemia growth *in vitro* and the pathogenetic role of plasma FST in leukemogenesis would have to be further investigated.

## Materials and Methods

### Zebrafish husbandry and mRNA microinjection

Wild-type AB zebrafish line (Zebrafish International Resource Center, ZIRC) was maintained under standard conditions (28.5°C, 14 h with light) in an automatically circulating system, and the embryos were kept in E3 medium (5 mM NaCl, 0.17 mM KCl, 0.33 mM CaCl$_2$, 0.33 mM MgSO$_4$) and staged as hour post-fertilization (hpf) and day post-fertilization (dpf) (Kimmel *et al*, 1995). *In vitro* synthesis of mRNA and microinjection were performed as described previously (He *et al*, 2014; Ma *et al*, 2017). Briefly, to overexpress *FLT3*/WT and *FLT3*/ITD in zebrafish embryos, full-length human *FLT3*/WT and *FLT3*/ITD sequences were cloned into pCS2 + plasmid, linearized by *NotI* digestion, purified (QIAquick PCR Purification Kit, Qiagen), and used as template for *in vitro* synthesis of capped mRNA (Ambion mMESSAGE mMACHINE™ SP6 Transcription Kit, Life Technologies). Synthesized mRNA was purified (RNeasy Mini Kit, Qiagen) and injected into one-cell stage embryos (300 pg per embryo). Embryos with severe developmental delay were excluded.

### Whole-mount *in situ* hybridization

Total RNA was isolated (TRIzol, Invitrogen) from 1-dpf-old AB zebrafish embryos and reverse-transcribed into cDNA (SuperScript II, Invitrogen). The latter was used as template to amplify (Ex Taq DNA polymerase, TaKaRa) the partial sequence of *fst* and *col9a2* (For primers, see Table EV4) by PCR. PCR products were cloned into a pGEM-T Easy Vector (Promega) and confirmed by directional sequencing (Centre for Genomic Sciences, The University of Hong Kong). The plasmid was linearized, purified, and used as template to generate the antisense RNA probe *in vitro* (DIG RNA Labeling Kit, Roche). Whole-mount *in situ* hybridization (WISH) was performed as described (He *et al*, 2014; Ma *et al*, 2017). On day 1, pre-fixed and permeabilized embryos were rehydrated, digested (embryos beyond 24 hpf) by proteinase K, re-fixed with 4% PFA, pre-hybridized in PHB- buffer (50% formamide, 5× SSC, 50 µg/ml heparin, 500 µg/ml yeast rRNA, 5 mM EDTA, 0.1% Tween-20, 0.92 mM citric acid, pH 6.0) at 65°C for 6 h, and hybridized in PHB+ buffer (PHB- with DIG-labeled RNA probe, 0.5 ng/µl) at 65°C overnight. After serial washing on day 2, the embryos were incubated with AP-conjugated anti-DIG antibody (1:5,000 in PBST with 5% lamb serum) at 4°C overnight with constant shaking. After serial washing on day 3, gene expression signals were developed by NBT/BCIP substrate (Roche), visualized, and captured by Nikon SMZ800 mounted with Nikon Digital Sight DS-Fil camera. WISH was performed in at least three biological triplicates. Twenty-five embryos were included in each experiment.

### Generation of Runx1-FLT3/ITD-transgenic zebrafish

Tol2-mediated transgenesis was used to generate Runx1-*FLT3*/ITD-transgenic zebrafish in which human *FLT3*/ITD was driven by

*Runx1* enhancer (Tamplin *et al*, 2015) and expressed in the hematopoietic stem/progenitor cells (HSPCs) (Suster *et al*, 2009). Briefly, human *FLT3*/ITD was cloned into the pDONR221 vector to generate the "middle" clone (pME-*FLT3*/ITD) by Gateway BP reaction. Three entry clones [p5E-Runx1 + 23 (Addgene #69602), pME-*FLT3*/ITD, and p3E-IRES-EGFPpA] were incorporated into destination vector pDestTol2pA by multisite Gateway LR reaction to generate pRunx1-*FLT3*/ITD-EGFP. pRunx1-*FLT3*/ITD-EGFP vector (50 pg) and transposase mRNA (50 pg) were co-injected into one-cell stage embryos, and those showing strong GFP expression at 4 dpf were raised to adulthood. F0 zebrafish were crossed with wild type to generate F1, which were genotyped by PCR of *GFP* and *FLT3* using gDNA from fin clip.

## Cell processing and molecular studies

Processing of primary AML sample and AML cell lines, real-time quantitative PCR (RT–qPCR), and Western blot analyses of zebrafish embryos, AML cell lines, normal PBSC, and primary AML have been described previously (He *et al*, 2014; Lam *et al*, 2016). AML cell lines, namely MV4-11, MOLM-13, THP-1, OCI-AML3, KG-1, ML2, NB-4, and Kasumi-1, are maintained and subcultured according to supplier's protocol (ATCC or DSMZ). All cell lines were recently authenticated and tested for mycoplasma contamination. Mononuclear cells in peripheral blood (PB) or bone marrow (BM) from AML patients were purified by standard density gradient centrifugation using Ficoll-Paque Plus Solution (GE Healthcare). Real-time quantitative PCR (RT–qPCR) was performed using SYBR Green reagents and StepOnePlus Real-Time PCR System (ABI). *β-Actin* and GAPDH were used as internal control for zebrafish and human samples, respectively. Fold change of gene of interest was calculated using the $2^{(-\Delta\Delta Ct)}$ method. Protein extracts from zebrafish, AML cell lines, and primary AML were fractionated in 10% SDS–PAGE gels and electrotransferred to nitrocellulose transfer membranes and blotted with primary antibody at 4°C overnight. After washing and incubating with HRP-conjugated secondary antibody at room temperature for 1 h, chemiluminescent substrate (Luminata Forte Western HRP Substrate, Millipore) will be applied and signals were visualized by ChemiDoc MP Imaging System (Bio-Rad Laboratories, Hercules, CA). Change in protein expression was quantified based on intensity of bands (Image Lab, Bio-Rad) as specified in the figure legends.

## Chromatin immunoprecipitation (ChIP)-qPCR

ChIP assay was performed as previously described with minor modifications (Sunadome *et al*, 2014). Briefly, cells were cross-linked by formaldehyde (1%) and quenched by glycine (0.125 M). Pellets were washed and resuspended in SDS lysis buffer with protease and phosphatase inhibitor. Chromatin was sonicated to obtain fragments of about 600 bp (Cole-Parmer Ultrasonic Processor), incubated with 2 μg rabbit normal IgG or anti-p-CREB (Ser133) antibody at 4°C overnight, enriched using protein A+G magnetic beads, washed by low-salt wash buffer, high-salt wash buffer, and LiCl wash buffer, and released using elution buffer at 65°C. DNA was treated with RNaseA (0.2 mg/ml) and proteinase K (0.2 mg/ml) and purified as template for quantitative PCR to amplify CREB-binding region in the *FST* promoter region. *c-Fos* was used as

positive control of p-CREB ChIP-qPCR. Compositions of various buffers and reagents, including dilution of each antibody, are shown in Table EV5.

## Establishment of inducible CRISPR/Cas9 system in human AML cell lines

MOLM-13 cell line was infected by pCW-Cas9, doxycycline-inducible lentiviral expressing Cas9 (Addgene plasmid #50661), and positively infected cells were selected by 2 μg/ml puromycin. sgRNAs (Table EV4) targeting p90RSK and CREB were cloned into pLKO5.sgRNA.EFS.tRFP657 (Addgene plasmid #57824). sgRNA lentivirus was prepared and used to infect inducible Cas9-expressing human AML cell lines. Positive sgRNA-infected cells were sorted based on tRFP657 expression. Cas9 was induced by 2 μg/ml doxycycline, and the cells were harvested 2 days after induction and subjected to subsequent analyses.

## Molecular cloning for FST overexpression, knockdown, and knockout

For overexpression, human *FST* isoforms (*FST317* and *FST344*) were cloned into the pLJM1-EGFP (Addgene #19319) lentivirus vector. For knockdown, two independent shRNAs were cloned into lentiviral vector pLVTHM (Addgene #12247). Scramble shRNA was used as control. For knockout, five different sgRNAs targeting exon 2 of human *FST* were designed (http://tools.genome-engineering.org) and cloned into pSpCas9(BB)-2A-GFP (Addgene #48138) followed by T7EI assay in 293FT. sgRNA#3 and sgRNA#4 showed highest genome-editing efficiency and were cloned into pL-CRISPR.EFS.GFP (Addgene #57818).

## Lentivirus packaging and transduction

Generation of viral particles and transduction in AML cell lines were performed as described previously (Man *et al*, 2012, 2014; Lam *et al*, 2016). Briefly, lentivirus was packaged in 293FT cells by co-transfecting lentiviral packaging plasmid pCMV-dR8.91, envelop plasmid pMD2.G, and transfer vectors for *FST* overexpression, knockdown, or knockout. Three days post-transfection, the supernatant was harvested, clarified through a 0.45-μm filter, and loaded in the sealing tubes (OptiSeal, Beckman) for ultracentrifugation (L-80, Beckman). Virus transduction in specified AML cell lines was performed by spinoculation in the presence of polybrene (8 μg/ml). Transduced cells were selected by puromycin (0.5 μg/ml) or FACS. Gene overexpression and knockdown efficiencies of transduced cells were confirmed by RT–qPCR and Western blotting. Knockout efficiencies by CRISPR/Cas9 were verified by T7EI assay as specified in the legends of appendix figure.

## RNA-sequencing and proteomics analysis

For RNA-sequencing, total RNA was extracted (RNeasy Mini Kit, Qiagen) from samples (ML-2-GFP, ML-2-FST317, ML-2-FST344) and the integrity was verified (Agilent 2100 bioanalyzer) before library construction and sequencing (Solexa HiSeq 1500). Sequencing reads were first filtered for adapter sequence, low-quality

sequence, and rRNA, and mapped to Human Genome GRCh38 (GENCODE) using STAR version 2.5.2. Gene expression was quantified by RSEM version 1.2.31, and differentially expressed (FDR < 0.05) transcripts were identified by EBSeq version 1.10.0 and visualized by volcano plot (GraphPad). Partek Genomics Suite and Gene Ontology Consortium (http://geneontology.org) were used for pathway and gene ontology enrichment analysis, respectively. For proteomics analysis, proteins were extracted from samples (ML-2-GFP, ML-2-FST317, ML-2-FST344) followed by trypsin digestion. LysC-tryptic peptides were cleaned and reconstituted in formic acid for LC-MS/MS analysis using an Orbitrap Fusion Lumos mass spectrometer interfaced with Dionex 3000RSLC nanoLC. High-resolution, high mass accuracy MS data were processed using Maxquant version 1.5.3.30, wherein MS data in triplicates for each condition were searched using the Andromeda algorithm against UniProt human protein databases as appropriate. Proteins identified from each experimental condition were quantified using the peptide LFQ intensities and compared based on changes of at least 1.5-fold. Data visualization and statistical data analysis were performed by Perseus software version 1.5.4.1. Gene Ontology Consortium (http://geneontology.org) was used for gene ontology enrichment analysis.

### Flt3/ITD knock-in mouse

The Flt3/ITD knock-in mouse line B6.129-Flt3$^{tm1Dgg}$/J was obtained from Jackson Laboratory. The genotype of Flt3 wild-type and ITD allele was determined by PCR using a common Flt3 forward primer (Flt3-F) with wild type (Flt3/WT-R) or ITD-specific reverse primer (Flt3/ITD-R; Table EV4) (Lee et al, 2007). To validate the expression of Flt3/ITD allele, total bone marrow cells were harvested and total RNA was extracted and reversely transcribed to first-stranded cDNA. Flt3_1952_forward primer and Flt3_2050_reverse primer (Table EV4) were used to amplify the juxtamembrane domain of mouse Flt3 to distinguish the expression of wild-type (99 bp) and ITD (117 bp) transcripts.

### Enzyme-linked immunosorbent assay (ELISA)

Peripheral blood from Flt3/ITD mice and their wild-type siblings was collected from the tail vein, clotted overnight at 4°C, and centrifuged at 1,000 g for 20 min at 4°C. The serum was collected and diluted to 10-fold for quantification by ELISA (Mouse FST ELISA Kit, LSBio #LS-F22235). Serum FST level from AML patients and AML cell line-xenografted NSG mice was measured by ELISA (Follistatin ELISA Kit, Abcam, ab192147). Briefly, serum samples or standards were incubated with antibody mix in each well and washed and bound FST was detected by standard colorimetric assays based on enzymatic reaction on TMB.

### Apoptosis and clonogenic analyses

Number of viable cells was enumerated by PrestoBlue assay (Lam et al, 2016). Apoptosis was performed using PE-conjugated Annexin V (BD Bioscience) and 7-aminoactinomycin D (7-AAD) according to the manufacturer's instructions. Clonogenicity was measured by standard methylcellulose-based system (MethoCult, Stem Cell Technologies). AML cells (100 cells for MOLM-13, 200 cells for MV4-11,

### The paper explained

#### Problem
Acute myeloid leukemia (AML) is an aggressive hematological malignancy with distinct cytogenetic, genetic, and clinicopathogenic features. Intensive chemotherapy and hematopoietic stem cell transplantation are the mainstays of treatment; however, treatment outcome is dismal. FLT3/ITD mutation occurs in about 30% of AML and is associated with poor prognosis. FLT3 inhibitors have been shown to be effective in FLT3/ITD AML; however, additional mutations of FLT3 confer drug resistance and treatment failure. It would be critical to identify new pathogenetic signals in FLT3/ITD AML.

#### Results
The present study demonstrates the following: (i) Ectopic expression of FLT3/ITD induces axis duplication in zebrafish embryos; (ii) the axis-duplicated embryos are associated with upregulation of a secreted morphogen FST; (iii) FST is a CREB target gene, which is consistently overexpressed in FLT3/ITD-expressed zebrafish, Flt3/ITD knock-in mice, and human FLT3/ITD AML; (iv) overexpression of FST potentiates MAPK/ERK signaling to promote leukemia cell growth; and (v) targeting FST reduces FLT3/ITD+ AML cell growth in vitro and in vivo.

#### Impact
This study suggests that FST is a novel therapeutic target and biomarker in human FLT3/ITD-mutated AML.

and 500 cells for ML-2) were seeded in triplicates in 35-mm culture plates. Colonies were stained with Giemsa (Sigma) and enumerated after 10 days of culture.

### Xenogeneic transplantation in NSG mice

Primary samples ($10 \times 10^6$ cells) or cell lines ($0.01 \times 10^6$ cells for MOLM-13 and $1 \times 10^6$ cells for ML-2) were suspended in 200 μl of Hanks' balanced salt solution and injected via tail vein into sublethally irradiated (205 cGy) 6- to 8-week-old male or female NSG mice (NOD.Cg-Prkdc$^{scid}$ Il2rg$^{tm1Wjl}$/SzJ, The Jackson Laboratory), which were housed and bred in the standard specific-pathogen-free (SPF) condition in Laboratory Animal Unit (LAU) of The University of Hong Kong. Engraftment was measured by whole animal bioluminescence or flow cytometry of BM aspirate at defined time points (Lam et al, 2016). For statistical analysis, 8–10 mice were used in each group for each xenogeneic transplantation experiment. For ASO treatment, MOLM-13-engrafted mice were randomly divided into two groups (Neg-ASO and FST-ASO3, five mice each) to minimize the effects of subjective bias.

### Study approval

All animal studies have been approved by the Committee on the Use of Live Animal for Teaching and Research at The University of Hong Kong and conformed to the ARRIVE guidelines. Human studies were approved by the appropriate Institutional Review Boards of Queen Mary Hospital, The University of Hong Kong. Informed consent was obtained from all subjects, and the experiments conformed to the principles set out in the WMA Declaration of Helsinki and the Department of Health and Human Services Belmont Report.

## Statistical analysis

Twenty-five zebrafish embryos at defined developmental stages were used in each group for whole-mount *in situ* hybridization, Western blotting, RT–qPCR, and drug treatment experiments. The number of mice and patient samples were specified in each figure legends. All experiments were performed in triplicates, and data were expressed as mean ± standard error of the mean (SEM) unless otherwise specified. Results were compared using the Student's *t*-test/Mann–Whitney *U*-test (numerical data) or Fisher's exact test (categorical data). Survival analysis was performed using the Kaplan–Meier method, and the differences in survival were determined using log-rank test. *P*-values less than 0.05, 0.01, and 0.001 were considered statistically significant and denoted by asterisk (*), (**), or (***), respectively. NS: not significant. The exact *P*-values are shown in a separate Supplemental Table (Appendix Table S1).

## Data availability

The RNA-seq data were deposited in NCBI GEO database (GEO accession: GSE115284; http://www.ncbi.nlm.nih.gov/geo/query/acc.cgi?acc=GSE115284).

**Expanded View** for this article is available online.

## Acknowledgements

We thank all patients for providing their blood and BM samples for this study. The zebrafish maintenance, PerkinElmer IVIS spectrum *in vivo* biolu-minescence imaging, RNA-seq, and proteomics analysis were supported by the Zebrafish Core Facility, Faculty Core Faculty, Centre for Genomic Science and Proteomics and Metabolomics Core Faculty, Li Ka Shing Faculty of Medicine, The University of Hong Kong. We thank Howard Chow, Harry Wong, Alan To, and Wan Liu for primary AML sample processing and data-base maintenance. We thank Dr. An-ming Meng (School of Life Science, Tsinghua University) for providing the −2,067gsc-gfp reporter plasmid and Dr. Koichi Kawakami (National Institute of Genetics, Japan) for the Tol2 kit. We thank Dr. Harvey Lodish (Department of Biology, MIT) for helpful comments for the manuscript. This project was supported by UGC-GRF (Ref no. 17126117) and Croucher Foundation. A.Y.H.L. was supported by Li Ka Shing Faculty of Medicine and by the endowment from the Li Shu Fan Medical Foundation.

## Author contributions

B-LH, NY, CHM, NK-LN, C-YC, S-SYL, C-WES, and AY-HL conducted study design, experiments, data acquisition, and analyses. B-LH, C-XZ, Y-LK, and AY-HL processed and analyzed the clinical samples and information. B-LH, H-CL, LL-HK, BY-LC, ML-LW, and AC-HM performed the zebrafish experi-ments. B-LH, NY, CHM, NK-LN, C-YC, and S-SYL performed the *in vitro* drug treatment and gene knockdown and overexpression experiments in AML cell lines. HK, GC, and RS conducted and analyzed the RNA-seq and proteomics analysis. Y-LK and AY-HL designed and conducted the clinical trial. BLHe wrote the manuscript. AY-HL and reviewed and agreed by all coauthors.

## Conflict of interest

The authors declare that they have no conflict of interest.

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
