## [Review Process File · EMBO Molecular Medicine]

Follistatin is a novel therapeutic target and biomarker in FLT3/ITD acute myeloid leukemia

Bai-Liang He, Ning Yang, Cheuk Him Man, Nelson Ka-Lam Ng, Chae-Yin Cher, Ho-Ching Leung, Leo Lai-Hok Kan, Bowie Yik-Ling Cheng, Stephen Sze-Yuen Lam, Michelle Lu-Lu Wang, Chun-Xiao Zhang, Hin Kwok, Grace Cheng, Rakesh Sharma, Alvin Chun-Hang Ma, Chi-Wai Eric So, Yok-Lam Kwong, and Anskar Yu-Hung Leung

Review timeline:

Submission date to The EMBO Journal:	8th May 2019
Editorial Decision:	15th May 2019
Transfer to EMBO Molecular Medicine:	15th May 2019
Editorial Decision:	18th Jun 2019
Revision received:	1st Jan 2020
Editorial Decision:	29th Jan 2020
Revision received:	7th Feb 2020
Accepted:	14th Feb 2020

Editor: Lise Roth

Transaction Report:

Editorial Decision from The EMBO Journal

15th May 2019

Thank you for submitting your manuscript "Follistatin is a novel therapeutic target and biomarker in FLT3/ITD acute myeloid leukemia" to The EMBO Journal. I have now read your study carefully and discussed the work with other members of the editorial team. I regret to inform you that we have decided not to pursue the publication at The EMBO Journal, but I recommend a transfer to EMBO Molecular Medicine, where the responsible editor Lise Roth would be happy to send the manuscript out for peer review.

We appreciate that the study describes activation of follistatin (FST) expression by AML-associated FLT3/ITD via the ERK-p90RSK-CREB pathway. The findings further show that FST promotes leukemia growth via induction of expression of genes linked to cancer initiation and progression. Furthermore, serum FST levels in FLT3/ITD AML patients and mice xenografted with patient-derived cancer cells correlate with the disease progression, and FST inhibition inhibits AML cell growth and prolongs survival of xenografted mice. While we appreciate the medical relevance of the identification of FST as a biomarker for FLT3/ITD-induced AML progression and potential treatment target, we also noted that follistatin has been shown to drive cancer progression in various cancer types, and its use as a cancer biomarker has been proposed. Additionally, FLT3 is known to activate MAPK pathway, and the role of MAPK and CREB in follistatin transcription regulation has been proposed in earlier work. Therefore, I am afraid we concluded that the provided broader conceptual advance of insight is not sufficient to consider publication at The EMBO Journal.

That being said, since I appreciate the more medical relevance of your study and the interest in identification of follistatin as a potential target in development of AML treatments, I have discussed the manuscript with my colleague Lise Roth at our sister journal EMBO Molecular Medicine, and she would be interested in sending your manuscript out for peer review. Should you be interested in transferring your manuscript to EMBO Molecular Medicine, please use the transfer link below (please note that no reformatting is required):

Thank you for the submission of your manuscript to EMBO Molecular Medicine, and please accept my apologies for the delay in getting back to you, which is due to the fact that one referee needed more time to complete his/her review. We have now received feedback from the three reviewers who agreed to evaluate your manuscript. As you will see from the reports below, the referees acknowledge the interest of the study. However, they also raise substantial concerns on your work, which should be convincingly addressed in a major revision of the present manuscript. In particular, it will be important to strengthen the proposed mechanism with rescue experiments and genetic loss-of-function to complement pharmacological inhibition. Moreover, the translational impact of the findings should be increased by showing the level of FST in patients' samples after relapse and adding *in vivo* models (primary xenotransplants with altered FST expression, use of immunocompetent mice). Attention should also be given to spelling and grammar.

Addressing the reviewers' concerns in full will be necessary for further considering the manuscript in our journal, and acceptance of the manuscript will entail a second round of review. EMBO Molecular Medicine encourages a single round of revision only and therefore, acceptance or rejection of the manuscript will depend on the completeness of your responses included in the next, final version of the manuscript. For this reason, and to save you from any frustrations in the end, I would strongly advise against returning an incomplete revision.

***** Reviewer's comments *****

Referee #1 (Comments on Novelty/Model System for Author):

The authors used various *in vitro* and *in vivo* methods to demonstrate that FST plays an important role in FLT3/ITD AML. This is the first report to show that FST may be a potential target in the treatment of FLT3/ITD AML.

Referee #1 (Remarks for Author):

In this study, the authors found axis duplication and dorsalization in zebrafish embryos expressed human FLT3/ITD which was accompanied by ectopic expression of a morphogen FST. With this finding, they explored the role of FST in FLT3/ITD AML and the mechanisms by *in vitro* and *in vivo* experiments. They found FST expression was also increased in FLT3/ITD knockin mice and human FLT3/ITD AML cells, compared with mobilized peripheral blood stem cells from healthy donors and FLT3/WT AML samples. Overexpression of human FST317 and FST344 isoforms enhanced leukemia cell growth while blocking FST had opposite effect. They showed direct binding of p-CREB to FST promoter in FLT3/ITD positive MOLM-13 by ChIP-PCR, and considered FST is a CREB target gene in FLT3/ITD AML. From the above findings and that inhibition of p90RSK, a known downstream effector of FLT3/ITD-ERK signaling cascade, reduced FST expression and phosphorylation of CREB, they suggested presence of a FLT3-p90RSK-CREB-FST signaling axis in FLT3/ITD AML. The findings from this study have potential clinical implication as FST may be a good target for the treatment of FLT3-ITD AML.

Major comment:

To say there is a FLT3-p90RSK-CREB-FST signaling axis in FLT3/ITD AML and FST is a CREB target gene, the authors need to show direct evidences. For example, can they demonstrate that deletion of the CREB binding site on FST will abolish the effect of FLT3/ITD or CREB expression? Similarly, whether knock down of FST will block FLT3/ITD induced expansion of myeloid and hematopoietic progenitor cells in zebrafish (shown in Figure 2)? What they showed are all indirect evidences. Without direct evidences, the statements had better be conservative.

Minor comment:

1. Page 6 line 129, Ectopic expression of *fst* (Figure 1M)....., 1M should be corrected to 1N.

2. The number of FLT3/WT AML patients in the study of serum FST level was too few though the FST level of the 3 patients studied was statistically lower than that of FLT3/ITD patients.

Referee #2 (Remarks for Author):

In the present manuscript, the authors report that hFLT3/ITD expression led to axis duplication and dorsalization in zebrafish embryos. The morphologic phenotype was accompanied by ectopic expression of follistatin (fst) during early embryonic development. The increase in fst expression also occurred in adult FLT3/ITD-transgenic zebrafish; Flt3/ITD knockin mice; and human FLT3/ITD AML cells. Overexpression of human FST317 and FST344 isoforms enhanced clonogenicity and leukemia engraftment in xenotransplantation model via RET, IL2RA, and CCL5 upregulation. FST shRNA, CRISPR/Cas9, or antisense oligo inhibited leukemic growth in vitro and in vivo. Serum FST positively correlated with leukemia engraftment in FLT3/ITD AML patient-derived xenograft mice and leukemia blast percentage in primary AML patients. In FLT3/ITD AML patients treated with FLT3 inhibitor quizartinib, serum FST levels correlated with clinical response.

The manuscript is interesting in particular for the biomarker potential of serum levels of FST in human.

It is to note that several manuscripts (often by the same authors) have previously pointed out (Blood 2017 130:4982; DOI: <https://doi.org/10.1016/j.exphem.2016.06.155>) the role of FST in Flt3/ITD leukemogenesis. Also the link with CREB has been reported previously (Blood 2017 130:4982).

Main points

-The authors show that the CREB inhibitor 666-15 (Kang, Lu et al., 2015) decreased FST expression (Figure 3L) and reduced cell growth of Ba/F3-FLT3/ITD without supplemental IL-3, but not Ba/F3 parental cells or Ba/F3-FLT3/ITD cells supplemented by IL-3 (Figure 3M). Furthermore, 666-15 partially rescued the morphologic anomalies induced by FLT3/ITD in zebrafish embryos (Figure 3N). Pharmacological inhibition should be implemented by using a phospho mutational approach.

-The authors suggest that the presence of a FLT3-p90RSK-CREB-FST signaling axis which might be relevant in the pathogenesis of FLT3/ITD. Would pharmacological/molecular unbalance of p90RSK rescue the anomalies?

- The authors suggest that FST is a potential therapeutic target for the treatment of FLT3/ITD AML. The in vivo experiments contrasting FST expression are mainly performed in cell lines (Fig. 6). The use of primary xenotransplants with altered FST expression should be included. In addition in these settings also the neutralizing antibody against FST might be used.

- The in vivo experiments are mainly performed in immunodeficient mice. Possibly different models (non immunocompromised) should be added.

- To identify the underlying molecular targets of FST, changes in gene expression profiles upon FST344 expression in ML-2 cells were examined. Since also FST317 was increasing leukemia engraftment, these experiments should also be shown as gene expression profiles.

-In fig. 7L correlation between serum FST levels and leukemia blast percentage from FLT3/ITD-mutated AML at diagnosis is shown. In Fig. 7M Serum FST decreased in CR and increased after relapse in 4 AML patients receiving Quizartinib monotherapy. These data are clinically very relevant and should possibly be implemented in number of patient's samples.

- Furthermore, in fig. 7I-K serum FST level was significantly increased in primary AML-derived xenografted mouse at week 6 post injection. Would the use of FST neutralizing antibody in this settings benefit against AML?

- How are the expression levels of IL2RA and CCL5 in the mouse AML and human AML settings ?

- Did the authors investigate the The source of follistatin in the serum from AML patients and AML mice? Is it derived from the blasts and if so is secretion important in disease development/maintenance?

-

Minor points

-The authors show that axis duplication and dorsalization were observed in 15.2 {plus minus} 1.3 and 34.7 {plus minus} 3.2 % of FLT3/ITD, but not FLT3/WT mRNA-injected embryos on 1 dpf (day post fertilization) (Figure 1A-D; Supplementary Figure 1A-J). The authors have previously shown that this phenotype was not observed in FLT3/ITD plasmid DNA-injected embryos (He et al., 2014). Why is this the case? Is it due to the different techniques applied or?

- Constitutive activation of FLT3 downstream signals STAT5, AKT and ERK were confirmed in zebrafish embryos (Figure 1J). Would the use of STAT5, AKT and ERK inhibitors reduce or block dorsalization?

- How was chordin (chd) expression?

Referee #3 (Comments on Novelty/Model System for Author):

Impressive breadth of model systems using zebrafish development and hematopoiesis, human cell lines in culture and in xenografts, and mouse knockin.

Referee #3 (Remarks for Author):

He et al.
EMBO Mol Med

The manuscript describes an upregulation in the expression of the morphogen Fst in zebrafish embryos and hematopoietic cells expressing the FLT3-ITD oncogene. The concept is novel and potentially significant in establishing the foundations for therapeutic or diagnostic applications. The data are clear, highlighted by the breadth of the study encompassing the analysis of developmental and hematopoietic phenotypes in zebrafish with followup analysis in mammalian cell lines and in vivo models. The observations of increased circulating FST in the FLT3-ITD knockin and in patients undergoing remission of FLT3-ITD AML support mechanistic observations made using cell lines. A few technical details which should be addressed are itemized below.

Figure 2M. Please indicate flow cytometry parameters on the axes of the plots.

Figure 3C. Quantitative data should be provided using qPCR amplification of ChIP samples.

Figure 3E. Indicate quantification data or adjust text to indicate "significant" increase.

Figure 3I - 3M. The analysis of the roles of RSK and CREB in mediating the effects of FLT3-ITD are based on inhibitor studies. These should be complemented with genetic loss-of-function, or at least using chemically distinct inhibitors with activity against these targets. The concern is that the inhibitors may affect additional targets beyond those identified in the manuscript.

Figure 4B. The increase in FST upon overexpression should be confirmed by western blot.

Supplementary Table 1 v. Supplementary Table 3. It appears that the correct table referenced at line 191 should be Supplementary Table 1. It is appreciated that the data are reported to be deposited in GEO.

Supplementary Figure 5. The text refers to these data as "cell growth" or proliferation, but the data are plotted as viability. What is the assay that is used in these experiments? It is not clear how the data support the conclusive statement in lines 218 -221. This discrepancy needs to be resolved.

Referee #1

Major comment:

1. To say there is a FLT3-p90RSK-CREB-FST signaling axis in FLT3/ITD AML and FST is a CREB target gene, the authors need to show direct evidences. For example, can they demonstrate that deletion of the CREB binding site on FST will abolish the effect of FLT3/ITD or CREB expression? Similarly, whether knock down of FST will block FLT3/ITD induced expansion of myeloid and hematopoietic progenitor cells in zebrafish (shown in Figure 2)? What they showed are all indirect evidences. Without direct evidences, the statements had better be conservative.

In the original manuscript, we demonstrated binding of p-CREB to FST promoter by ChIP-PCR. To provide direct evidence as suggested by this reviewer, we have performed dual luciferase reporter assay and demonstrated that deletion of CREB binding site on FST promoter abolished the effect of CREB-mediated FST up-regulation. Briefly, wildtype (CRE+) or CREB binding site-deleted (CRE-) FST promoter were cloned into Luciferase Reporter Vectors (pGL3-Basic, Promega) and co-transfected with constitutively active form of CREB_{Y134F} (pLV-CREB_{1Y134F}-GFPSpark plasmid) and control plasmid. The pRL-CMV vector was used as Renilla Luciferase Control Reporter. Deletion of CREB binding site significantly reduced the relative light unit (RLU, Firefly / Renilla Luciferase). Knockdown of *fst* in zebrafish resulted in severe perturbation of dorsoventral patterning during early embryonic development and thus the effects on FLT3-ITD expression on myelopoiesis could not be ascertained. The manuscript was revised as follows:

Page 8, Line 155:

Direct binding of p-CREB to FST promoter in FLT3/ITD positive MOLM-13 cells was confirmed by ChIP-qPCR. Compared with normal IgG control, there was a 6-fold increase in DNA binding to p-CREB in FST promoter (Figure 3B and 3C). Dual luciferase reporter assay demonstrated that deletion of CREB binding site on FST promoter abolished the effect of CREB-mediated FST up-regulation (Figure 3D).

Page 19, Line 399:

Chromatin immunoprecipitation (ChIP)-qPCR

ChIP assay was performed as previously described with minor modifications (Sunadome et al, 2014). Briefly, cells were cross-linked by formaldehyde (1%) and quenched by glycine (0.125 M). Pellets were washed and resuspended in SDS lysis buffer with protease and phosphatase inhibitor. Chromatin was sonicated to obtain fragments of 600 bp (Cole-Parmer Ultrasonic Processor), incubated with 2 µg rabbit normal IgG or anti-p-CREB (Ser133) antibody at 4°C overnight, enriched using protein A+G magnetic beads, washed by low salt wash buffer, high salt wash buffer and LiCl wash buffer, and released using elution buffer at 65°C. DNA was treated with RNaseA (0.2 mg/mL) and proteinase K (0.2mg/mL) and purified as template for quantitative PCR to amplify CREB binding region in the FST promoter region. c-Fos was used as positive control of p-CREB ChIP-qPCR. Compositions of various buffers and reagents were shown in Appendix Table S5.

Page 42, Line 853:

Legends for Figure 3. B-C *The direct binding of p-CREB to human FST promoter was detected by ChIP-PCR (B) and ChIP-qPCR. c-Fos was used as positive control of p-CREB target gene. Normal IgG was used as negative control of ChIP. D* **Dual luciferase assay demonstrating the direct binding of pCREB on human FST promoter.** *pRL-CMV, Renilla luciferase vector; pGL-CRE- and pGL-CRE+, firefly luciferase expression driven by human FST promoter with deleted CRE site (CRE-) or wildtype (CRE+); p-GFPSpark, GFP expressing vector; p-CREB_{Y134F}, CREB_{Y134F}-GFP expressing vector.*

Minor comment:

1. Page 6 line 129, Ectopic expression of *fst* (Figure 1M)....., 1M should be corrected to 1N.

The manuscript has been revised.

Page 6, Line 129:

*Ectopic expression of *fst* (Figure 1N) and gooseoid (*gsc*) was also shown by WISH (Appendix Figure S1K) and GFP reporter assay (Appendix Figure SIL-N).*

2. The number of FLT3/WT AML patients in the study of serum FST level was too few though the FST level of the 3 patients studied was statistically lower than that of FLT3/ITD patients.

In the original manuscript, we demonstrated that cellular FST in FLT3-ITD AML was significantly higher than that in FLT3-WT AML at both transcript and protein levels. However, serum FST levels in different AML patients varied widely and there was no convincing difference between FLT3-ITD and FLT3-WT AML. FST has been shown to express in different tissues (Phillips and de Kretser 1998) and we reckoned that serum FST could be contributed by tissues other than myeloblasts, making comparison between individual patients difficult. The manuscript has been revised as follows:

Page 4, Line 73:

FST has been shown to express in different tissues (Phillips & de Kretser, 1998) and antagonizes Activin A, a member of the TGF- β (Transforming Growth Factor- β) superfamily (Cash et al, 2012).

Page 13, Line 281:

However, serum FST levels in different AML patients varied widely and there was no convincing difference between FLT3-ITD and FLT3-WT AML (Appendix Figure S11).

A new reference was added:

Phillips DJ, de Kretser DM (1998) Follistatin: a multifunctional regulatory protein. *Front Neuroendocrinol* 19: 287-322

Referee #2 (Remarks for Author):

1. The manuscript is interesting in particular for the biomarker potential of serum levels of FST in human. It is to note that several manuscripts (often by the same authors) have previously pointed out (Blood 2017 130:4982; DOI: <https://doi.org/10.1016/j.exphem.2016.06.155>) the role of FST in Flt3/ITD leukemogenesis. Also the link with CREB has been reported previously (Blood 2017 130:4982).

The reports mentioned were abstracts of initial findings that we presented in international conferences about the role of FST in FLT3-ITD AML.

Main points

2. The authors show that the CREB inhibitor 666-15 (Kang, Lu et al., 2015) decreased FST expression (Figure 3L) and reduced cell growth of Ba/F3-FLT3/ITD without supplemental IL-3, but not Ba/F3 parental cells or Ba/F3-FLT3/ITD cells supplemented by IL-3 (Figure 3M). Furthermore, 666-15 partially rescued the morphologic anomalies induced by FLT3/ITD in zebrafish embryos (Figure 3N). Pharmacological inhibition should be implemented by using a phospho mutational approach.

In the original manuscript we made use of CREB inhibitor 666-15 and demonstrated a FLT3-p90RSK-CREB-FST signaling axis in FLT3-ITD AML. In the revised manuscript, CRISPR/Cas9 system was developed in which sgRNAs targeting CREB were transduced into MOLM-13 cell line carrying inducible Cas9. Upon Cas9 induction by doxycycline exposure (2 μ g/mL, 2 days), CREB knockout was confirmed and FST expression was significantly reduced. Furthermore, we have performed dual luciferase assay, showing that mutant form of CREB induced significant increase in FST promoter activities. The observations strengthened the proposition that CREB activated FST expression. The manuscript was revised as follows:

Page 8, Line 158:

Dual luciferase reporter assay demonstrated that deletion of CREB binding site on FST promoter abolished the effect of CREB-mediated FST up-regulation (Figure 3D).

Page 8, Line 174:

Specifically, CREB inhibitor 666-15 (Kang et al, 2015) or knockout by CRISPR/Cas9 also decreased FST expression (Figure 3M and 3N) and 666-15 reduced cell growth of Ba/F3-FLT3/ITD without supplemental IL-3, but not Ba/F3 parental cells or Ba/F3-FLT3/ITD cells supplemented by IL-3 (Figure 3O).

Page 15, Line 325:

p90RSK and CREB knockout by CRISPR/Cas 9 also induced significant decrease in FST expression.

Page 19, Line 399:

Chromatin immunoprecipitation (ChIP)-qPCR

ChIP assay was performed as previously described with minor modifications (Sunadome et al, 2014). Briefly, cells were cross-linked by formaldehyde (1%) and quenched by glycine (0.125 M). Pellets were washed and resuspended in SDS lysis buffer with protease and phosphatase inhibitor. Chromatin was sonicated to obtain fragments of 600 bp (Cole-Parmer Ultrasonic Processor), incubated with 2 µg rabbit normal IgG or anti-p-CREB (Ser133) antibody at 4°C overnight, enriched using protein A+G magnetic beads, washed by low salt wash buffer, high salt wash buffer and LiCl wash buffer, and released using elution buffer at 65°C. DNA was treated with RNaseA (0.2 mg/mL) and proteinase K (0.2mg/mL) and purified as template for quantitative PCR to amplify CREB binding region in the FST promoter region. c-Fos was used as positive control of p-CREB ChIP-qPCR. Compositions of various buffers and reagents were shown in Appendix Table S5.

Page 19, Line 413:

Establishment of inducible CRISPR/Cas9 system in human AML cell lines

MOLM-13 cell line was infected by pCW-Cas9, doxycycline-inducible lentiviral expressing Cas9 (Addgene plasmid #50661) and positively infected cells were selected by 2µg/ml puromycin. sgRNAs (Appendix Table S4) targeting p90RSK and CREB were cloned into pLKO5.sgRNA.EFS.tRFP657 (Addgene plasmid #57824). sgRNA lentivirus was prepared and used to infect inducible Cas9 expressing human AML cell lines. Positive sgRNA infected cells were sorted based on tRFP657 expression. Cas9 was induced by 2µg/ml doxycycline and the cells were harvested 2 days after induction and subjected to subsequent analyses.

Page 42, Line 853:

Legends for Figure 3. B-C The direct binding of p-CREB to human FST promoter was detected by ChIP-PCR (B) and ChIP-qPCR. c-Fos was used as positive control of p-CREB target gene. Normal IgG was used as negative control of ChIP. **D** Dual luciferase assay demonstrating the direct binding of pCREB on human FST promoter. pRL-CMV, Renilla luciferase vector; pGL-CRE- and pGL-CRE+, firefly luciferase expression driven by human FST promoter with deleted CRE site (CRE-) or wildtype (CRE+); p-GFPspark, GFP expressing vector; p-CREBY134F, CREBY134F-GFP expressing vector.

Page 43, Line 874:

Legends to Figure 3. N CREB and FST expression were detected by Western Blotting after CREB knockout by CRISPR/Cas9 in MOLM-13 cells.

3. The authors suggest that the presence of a FLT3-p90RSK-CREB-FST signaling axis which might be relevant in the pathogenesis of FLT3/ITD. Would pharmacological/molecular unbalance of p90RSK rescue the anomalies?

In the previous manuscript, we demonstrated that p90RSK inhibitor BRD7389 reduced FST expression and cell growth of FLT3-ITD AML. We have designed CRISPR/Cas9 knockout of p90RSK and showed that it significantly reduced FST expression in FLT3/ITD+ MOLM-13 AML cells. The revised manuscript should now read:

Page 8, Line 171:

Consistently, a doxycycline inducible system showed that knockout of p90RSK by CRISPR/Cas9 resulted in significant decrease in FST expression in MOLM-13 cell line (Figure 3L).

Page 15, Line 325:

p90RSK and CREB knockout by CRISPR/Cas 9 also induced significant decrease in FST expression.

Page 19, Line 413:

Establishment of inducible CRISPR/Cas9 system in human AML cell lines

MOLM-13 cell line was infected by pCW-Cas9, doxycycline-inducible lentiviral expressing Cas9 (Addgene plasmid #50661) and positively infected cells were selected by 2ug/ml puromycin. sgRNAs (Appendix Table S4) targeting p90RSK and CREB were cloned into pLKO5.sgRNA.EFS.tRFP657 (Addgene plasmid #57824). sgRNA lentivirus was prepared and used to infect inducible Cas9 expressing human AML cell lines. Positive sgRNA infected cells were sorted based on tRFP657 expression. Cas9 was induced by 2ug/ml doxycycline, cell upon gene knockout was harvested 2 days after induction and subjected to subsequent analyses.

Page 42, Line 870:

Legends to Figure 3. L RSK and FST expression were detected by Western Blotting after p90RSK knockout by CRISPR/Cas9 in MOLM-13 cells.

4. The authors suggest that FST is a potential therapeutic target for the treatment of FLT3/ITD AML. The in vivo experiments contrasting FST expression are mainly performed in cell lines (Fig. 6). The use of primary xenotransplants with altered FST expression should be included. In addition in these settings also the neutralizing antibody against FST might be used.

To demonstrate that FST is a potential therapeutic target for the treatment of FLT3/ITD AML, we made use of FLT3/ITD+ MOLM-13 cell line and showed that FST knock-down by shRNA (Fig. 5) or FST-specific antisense oligo (Fig. 6F-H) as well as knock-out by CRISPR/Cas9 (Fig. 6A-E) all resulted in significant reduction of leukemia growth both *in vitro* and *in vivo*. The cell line model provided a consistent model with high gene transduction efficiency, hence the effects of FST perturbation could be ascertained. FST neutralizing antibody had no effect on leukemia growth *in vitro* (Appendix Figure S7E) and hence it was not tested *in vivo*.

We also made use of primary FLT3-ITD AML patients-derived xenografts in NSG mice to examine the link between FLT3-ITD AML development and serum FST. In this way, we compared serum FST in mice before and after leukemia engraftment within each AML sample to avoid variation between samples and demonstrated a significant increase in FST level 6-week post transplantation (Figure 7K).

5. The in vivo experiments are mainly performed in immunodeficient mice. Possibly different models (non immunocompromised) should be added.

Xenotransplantation in immunodeficient mouse model e.g. NSG mice is the gold standard for the enumeration of human leukemia activity *in vivo* and that was used throughout the study. We agreed with this reviewer that different models should be considered and have made use of FLT3/ITD knock-in mouse model in which we demonstrated a significant increase in serum FST compared with their wildtype siblings (Figure 7A-D).

6. To identify the underlying molecular targets of FST, changes in gene expression profiles upon FST344 expression in ML-2 cells were examined. Since also FST317 was increasing leukemia engraftment, these experiments should also be shown as gene expression profiles.

In the original manuscript, we performed transcriptome analysis of *FST344* expression (the predominant FST form) in ML-2 cell line, in which endogenous FST expression was minimal. In the revised manuscript, we expressed *FST317* in ML-2 and performed similar analysis. The transcriptome profiles resulting from the expression of two FST forms were quite distinct though some of the differentially expressed genes have been associated with inferior treatment outcome in AML. The pathogenetic significance of these findings is currently unclear. The revised manuscript should now read:

Page 10, Line 209:

RNA-seq was also performed in ML-2 cells-overexpressed *FST317* comparing to those with GFP overexpression. Intriguingly, the transcriptome profile induced by *FST317* overexpression was distinct from those of *FST344* overexpression (Appendix Figure S5A, Appendix Table S2). Specifically, the differentially expressed gene RAS oncogene family like 6 (*RABL6*, also known as *RBEL1* and *C9orf86*) (Hagen et al, 2014), *CD93* (Iwasaki et al, 2015), and Zinc Finger Protein 709

(ZNF709) (Yan et al, 2016) have been implicated in cancers and leukemia. Upregulation of *PRTFDC1* and *PODXL2* expression were correlated with poor survival of AML whereas down-regulation of *CCNL1* and *RP11-762I7.5* were correlated with poor survival of AML (Appendix Figure S5B).

Figure S5 and legends were added in the Appendix.

7. In fig. 7L correlation between serum FST levels and leukemia blast percentage from FLT3/ITD-mutated AML at diagnosis is shown. In Fig. 7M Serum FST decreased in CR and increased after relapse in 4 AML patients receiving Quizartinib monotherapy. These data are clinically very relevant and should possibly be implemented in number of patient's samples.

We agreed with this reviewer. Quizartinib (or gilteritinib) has not been approved for clinical application in Hong Kong and the access to quizartinib has been limited to those who were recruited into clinical trial which has been completed. Therefore, we were unable to increase sample numbers for the time being. The point has been incorporated to the revised manuscript that should now read:

Page 47, Line 940:

Legend to Figure 7. *N* Serum FST continued to rise during disease progression from a patient who did not respond to Quizartinib. Patients in *M* and *N* were recruited in the QUANTUM-R and patient accrual has been completed.

8. Furthermore, in fig. 7I-K serum FST level was significantly increased in primary AML-derived xenografted mouse at week 6 post injection. Would the use of FST neutralizing antibody in this settings benefit against AML?

Intriguingly, FST neutralizing antibody had no effects *in vitro* despite high concentration (Appendix Fig. S7E), in contrast to FST knockdown by anti-sense oligo, which has been shown to reduce leukemia growth both *in vitro* and *in vivo* (Fig. 6F-H). We speculated that while serum FST might reflect AML disease activity, it was the intracellular FST that presented potential target for therapeutic intervention. These points were incorporated into the revised manuscript as follows:

Page 17, Line 372:

Intriguingly, FST neutralizing antibodies had no significant effect on leukemia growth in vitro and the pathogenetic role of plasma FST in leukemogenesis would have to be further investigated.

9. How are the expression levels of IL2RA and CCL5 in the mouse AML and human AML settings?

In the original manuscript, we demonstrated, based on public databases, the inferior treatment outcome of IL2RA and CCL5 expression in human AML. In the revised manuscript, we looked up databases in public domains and realized that the expression levels of IL2RA and CCL5 were significantly higher in AML than normal hematopoietic tissues. The revised manuscript should now read:

Page 9, Line 198:

Upregulation of RET, IL2RA, and CCL5 were confirmed by RT-qPCR (Fig. 4J), of which IL2RA and CCL5 were shown to highly express in human AML compared with normal hematopoietic tissues (Appendix Figure S4) and were associated with poor overall survival in AML patients (Figure 4K and L).

Page 15, Line 329:

Specifically, FST overexpression in ML-2 cells was shown to upregulate genes involved in MAPK/ERK pathway including RET, IL2RA, and CCL5; of which IL2RA and CCL5 were shown to highly express in human AML compared with normal hematopoietic tissues (Bloodspot website: <http://servers.binf.ku.dk/bloodspot/>) and were associated with poor overall survival in AML patients.

A new Figure S4 was added in the Appendix.

10. Did the authors investigate the source of follistatin in the serum from AML patients and AML mice? Is it derived from the blasts and if so is secretion important in disease development/maintenance?

In the original manuscript, we demonstrated an increase in serum FST in NSG mice upon engraftment by MOLM-13 cell line or primary human FLT3-ITD AML cells. Furthermore, in patients with FLT3-ITD AML who were treated with quizartinib, serum FST was correlated with disease status. These observations suggested that leukemic blasts would at least partially contribute to serum FST. Intriguingly, FST neutralizing antibody had no effects *in vitro* despite high concentration (Appendix Figure S5E), in contrast to FST knockdown by anti-sense oligo, which has been shown to reduce leukemia growth both *in vitro* and *in vivo* (Fig. 6F-H). We speculated that while serum FST might reflect AML disease activity, it was intracellular FST that presented a target for therapeutic intervention. These points were incorporated into the revised manuscript as follows:

Page 17, Line 369:

Despite its ubiquitous tissue expression, the observations aforementioned supported the proposition that FLT3-ITD AML blasts contributed, at least partially, to plasma FST. Intriguingly, FST neutralizing antibodies had no significant effect on leukemia growth in vitro and the pathogenetic role of plasma FST in leukemogenesis would have to be further investigated.

Minor points

1. The authors show that axis duplication and dorsalization were observed in 15.2 {plus minus} 1.3 and 34.7 {plus minus} 3.2 % of FLT3/ITD, but not FLT3/WT mRNA-injected embryos on 1 dpf (day post fertilization) (Figure 1A-D; Supplementary Figure 1A-J). The authors have previously shown that this phenotype was not observed in FLT3/ITD plasmid DNA-injected embryos (He et al., 2014). Why is this the case? Is it due to the different techniques applied or?

The apparent discrepancies in morphologic phenotypes were related to the difference in temporal and spatial expression of nucleic acids injected. In our previous study, overexpression of FLT3/ITD by plasmid DNA injection resulted in relatively late (after 6 hours post fertilization) and mosaic expression of FLT3/ITD that had minimal effects on dorsal-ventral patterning. In the present study, injection of FLT3/ITD mRNA into one-cell stage zebrafish embryos resulted in early and ubiquitous expression of FLT3/ITD proteins that induced axis duplication and dorsalization in embryos. These points are incorporated into the revised manuscript, which should now read:

Page 14, Line 288:

The serendipitous axis-duplication and dorsalization in zebrafish embryos were not detected in our previous studies (He et al., 2014). The apparent discrepancies in morphologic phenotypes were related to the difference in temporal and spatial expression of nucleic acids injected. In our previous study, overexpression of FLT3/ITD by plasmid DNA injection resulted in relatively late (after 6 hours post fertilization) and mosaic expression of FLT3/ITD that had minimal effects on dorsal-ventral patterning. In the present study, injection of FLT3/ITD mRNA into one-cell stage zebrafish embryos resulted in early and ubiquitous expression of FLT3/ITD proteins that induced axis duplication and dorsalization in embryos.

2. Constitutive activation of FLT3 downstream signals STAT5, AKT and ERK were confirmed in zebrafish embryos (Figure 1J). Would the use of STAT5, AKT and ERK inhibitors reduce or block dorsalization?

The use of STAT5, AKT and ERK inhibitors were associated with significant embryonic toxicities, making it difficult to ascertain effects on dorsalization or axis duplication induced by FLT3-ITD expression.

3. How was chordin (chd) expression?

We have examined Chordin (chd) expression (RT-qPCR) at 6 hours post fertilization in FLT3/WT and FLT3/ITD mRNA-injection embryos. There was no significant difference in these embryos and

the results were shown in Figure 1L.

Referee #3 (Remarks for Author):

1. Figure 2M. Please indicate flow cytometry parameters on the axes of the plots.

We apologize for the oversight. The axes in Figure 2M have been labeled in the revised Figure 2M.

2. Figure 3C. Quantitative data should be provided using qPCR amplification of ChIP samples.

In the original manuscript, intensity of FST PCR product as shown in Figure 3D was quantified by ImageJ. In the revised manuscript, we performed quantitative PCR to provide more accurate evaluation of CREB binding to FST promoter. Chromatin immune-precipitation was performed using anti-p-CREB antibody. Normal IgG was used as a negative control. Quantitative PCR was used to amplify CREB binding region of FST and c-FOS promoters. The latter is a known CREB target and was used as positive control. Compared with normal IgG control, there was a 6-fold increase in DNA binding to p-CREB in FST promoter. The revised manuscript should now read:

Page 8, Line 156:

Compared with normal IgG control, there was a 6-fold increase in DNA binding to p-CREB in FST promoter (Figure 3B and 3C).

Page 19, Line 407:

DNA was treated with RNaseA (0.2 mg/mL) and proteinase K (0.2mg/mL) and purified as template for quantitative PCR to amplify CREB binding region in the FST promoter region.

Page 42, Line 853:

Legend to Figure 3. B-C *The direct binding of p-CREB to human FST promoter was detected by ChIP-PCR (B) and ChIP-qPCR. c-Fos was used as positive control of p-CREB target gene. Normal IgG was used as negative control of ChIP.*

3. Figure 3E. Indicate quantification data or adjust text to indicate "significant" increase.

The manuscript has been revised which should now read:

Page 8, Line 161:

Consistently, FLT3/ITD, its downstream signals including p-STAT5, p-ERK1/2, p-AKT and p-4E-BP1 and FST expression were increased in Ba/F3-FLT3/ITD cells compared to the parental cells (Figure 3E).

4. Figure 3I - 3M. The analysis of the roles of RSK and CREB in mediating the effects of FLT3-ITD are based on inhibitor studies. These should be complemented with genetic loss-of-function, or at least using chemically distinct inhibitors with activity against these targets. The concern is that the inhibitors may affect additional targets beyond those identified in the manuscript.

In the original manuscript we made use of p90RSK inhibitor BRD7389 and CREB inhibitor 666-15 and demonstrated a FLT3-p90RSK-CREB-FST signaling axis in FLT3-ITD AML. In the revised manuscript, we interrogated the axis by both gene knockout and pharmacologic inhibition and confirmed the pathogenetic role of this axis. Specifically, CRISPR/Cas9 system was developed in which sgRNAs targeting p90RSK and CREB were transduced into MOLM-13 cell line carrying inducible Cas9. Upon Cas9 induction by doxycycline exposure, p90RSK and CREB knockout was confirmed and FST expression was significantly reduced.

Page 8, Line 171:

Consistently, a doxycycline inducible system showed that knockout of p90RSK by CRISPR/Cas9 resulted in significant decrease in FST expression in MOLM-13 cell line (Figure 3L). Specifically, CREB inhibitor 666-15 (Kang et al, 2015) or knockout by CRISPR/Cas9 also decreased FST expression (Figure 3M and 3N) and 666-15 reduced cell growth of Ba/F3-FLT3/ITD without supplemental IL-3, but not Ba/F3 parental cells or Ba/F3-FLT3/ITD cells supplemented by IL-3 (Figure 3O). Furthermore, 666-15 treatment partially rescued the morphologic anomalies induced by FLT3/ITD in zebrafish embryos (Figure 3P) without observable toxicity, underscoring the preferential role of CREB in FLT3/ITD signaling. These observations demonstrated the presence of a FLT3-p90RSK-CREB-FST signaling axis that might be relevant in the pathogenesis of FLT3/ITD.

Page 15, Line 325:

p90RSK and CREB knockout by CRISPR/Cas 9 also induced significant decrease in FST expression.

Page 19, Line 413:

Establishment of inducible CRISPR/Cas9 system in human AML cell lines

MOLM-13 cell line was infected by pCW-Cas9, doxycycline-inducible lentiviral expressing Cas9 (Addgene plasmid #50661) and positively infected cells were selected by 2ug/ml puromycin. sgRNAs (Appendix Table S4) targeting p90RSK and CREB were cloned into pLKO5.sgRNA.EFS.tRFP657 (Addgene plasmid #57824). sgRNA lentivirus was prepared and used to infect inducible Cas9 expressing human AML cell lines. Positive sgRNA infected cells were sorted based on tRFP657 expression. Cas9 was induced by 2ug/ml doxycycline and cells were harvested 2 days after induction and subjected to subsequent analyses.

Page 42, Line 870:

Legends to Figure 3. *L RSK and FST expression were detected by Western Blotting after p90RSK knockout by CRISPR/Cas9 in MOLM-13 cells. M The phosphorylation of CREB and FST expression were detected by Western Blotting in Ba/F3-FLT3/ITD cells treated with CREB inhibitor 666-15 for one day. ^: non-specific staining of p-ATF1 protein due to the conserved motif. N CREB and FST expression were detected by Western Blotting after CREB knockout by CRISPR/Cas9 in MOLM-13 cells.*

5. Figure 4B. The increase in FST upon overexpression should be confirmed by western blot.

We have performed Western Blot for FST in ML-2 cell line, in which FST317 and FST344 were over-expressed, and confirmed that FST protein expression particularly the FST344 form was significantly increased. The results were shown in the revised Figure 4B. The manuscript was revised as follows:

Page 9, Line 187:

Overexpression of the 2 spliced forms of FST (FST317 and FST344) was confirmed at mRNA and protein levels (Figure 4B) and was shown to enhance cell growth (Figure 4C) and clonogenicity (Figure 4D and E) in vitro.

Page 44, Line 882:

Legend to Figure 4. B-C FST317 and FST344 overexpression resulted in significant increases of FST transcription by RT-qPCR and protein by Western Blot.

6. Supplementary Table 1 v. Supplementary Table 3. It appears that the correct table referenced at line 191 should be Supplementary Table 1. It is appreciated that the data are reported to be deposited in GEO.

We apologize for the oversight. The manuscript has been revised. The data has been deposited in GEO.

7. Supplementary Figure 5. The text refers to these data as "cell growth" or proliferation, but the data are plotted as viability. What is the assay that is used in these experiments? It is not clear how the data support the conclusive statement in lines 218 -221. This discrepancy needs to resolved.

In the new Appendix Figure S7 (corresponding to Supplementary Figure 5 of previous version), viable cells were numerated by PrestoBlue assay that provided information about number of viable cells (Lam et al., 2016). The manuscript has been revised which should now read:

Page 11, Line 233:

However, exogenous FST (Appendix Figure S7B) and Activin A (Appendix Figure S7C) as well as Activin receptor antagonist (Appendix Figure S7D) and FST neutralizing antibody (Appendix Figure S7E) had no significant effect on MOLM-13 cell growth in vitro, enumerated as the number of viable cells based on PrestoBlue assay.

Page 22, Line 474:

Number of viable cells was enumerated by PrestoBlue assay (Lam et al., 2016).

Comment from Editor Dr. Lise Roth

In particular, it will be important to strengthen the proposed mechanism with rescue experiments and genetic loss-of-function to complement pharmacological inhibition. Moreover, the translational impact of the findings should be increased by showing the level of FST in patients' samples after relapse and adding *in vivo* models (primary xenotransplants with altered FST expression, use of immunocompetent mice). Attention should also be given to spelling and grammar.

We are grateful to the comments from the reviewers, which have been insightful and constructive. Based on these comments, we have performed new experiments and the results have strengthened the proposed FLT3-p90RSK-CREB-FST signaling axis. Specifically, dual luciferase promoter assay further confirmed direct binding of CREB to FST promoter. Knockout of p90RSK and CREB using CRISPR/Cas9 has again delineated their pathogenetic role in FST induction. Xenotransplantation of primary AML samples has been performed and the significant increase in serum FST level upon leukemia engraftment has underscored the relevance of serum FST as biomarkers of disease activity. Immunocompetent Flt3-ITD knockin mouse model has also confirmed the pathogenetic link between FLT3-ITD and FST level. The revised manuscript has been proof-read by a native English writer to ensure grammar and spelling were corrected.

3rd Editorial Decision

29th Jan 2020

Thank you for the submission of your revised manuscript to EMBO Molecular Medicine. We have now received the enclosed report from the referees who were asked to re-assess it. As you will see, they are overall supportive of publication, and I am thus pleased to inform you that we will be able to accept your manuscript pending the following final editorial amendments.

***** Reviewer's comments *****

Referee #1 (Remarks for Author):

The authors have responded to my comments and shown direct evidence that FST is a CREB target gene in FLT3/ITD AML.
I have no more comments.

Referee #2 (Comments on Novelty/Model System for Author):

adequate

Referee #2 (Remarks for Author):

The present revised version of the manuscript is implemented and the medical value is much more clear. The authors have included additional experiments where requested or have given a plausible explanation.

As a minor comment to the previous question:

The fact that the use of STAT5, AKT and ERK inhibitors were associated with significant embryonic toxicities, making it difficult to ascertain effects on dorsalization or axis duplication induced by FLT3-ITD expression... should be included in the discussion, indicating that experiments have been carried out and the reason why the outcome is of difficult interpretation.

Referee #3 (Comments on Novelty/Model System for Author):

The technical quality of the paper is appreciated in the investigation of autochthonous models of FLT3-ITD leukemia, xenograft models, and human patient responses to FLT3-ITD therapy. These studies firmly establish the relationship between FLT3-ITD and FST. The mechanistic investigation of RSK1 in the regulation of FST is adequate given the overall advancement linking FLT3-ITD to FLT. The finding seems to be completely novel. the medical impact from the human patient response data is clear.

Referee #3 (Remarks for Author):

The revised manuscript has thoroughly addressed previous points.

Remaining comments:

In lines 169 and following, please indicate the RSK paralog that is studied, namely p90RSK1.

2nd Revision - authors' response

7th Feb 2020

The authors performed the requested editorial changes.

Corresponding Author Name: Anskar Yu-Hung Leung

Manuscript Number: EMM-2019-10895-V2